# A Review for Uncovering the “Protein-Nanoparticle Alliance”: Implications of the Protein Corona for Biomedical Applications

**DOI:** 10.3390/nano14100823

**Published:** 2024-05-08

**Authors:** Burcu Önal Acet, Désirée Gül, Roland H. Stauber, Mehmet Odabaşı, Ömür Acet

**Affiliations:** 1Faculty of Arts and Science, Chemistry Department, Aksaray University, Aksaray 68100, Turkey; burcuonal@asu.edu.tr (B.Ö.A.); modabasi@aksaray.edu.tr (M.O.); 2Department of Otorhinolaryngology Head and Neck Surgery, Molecular and Cellular Oncology, University Medical Center, 55131 Mainz, Germany; rstauber@uni-mainz.de; 3Vocational School of Health Science, Pharmacy Services Program, Tarsus University, Tarsus 33100, Turkey

**Keywords:** protein corona, nanoparticle, nanoparticle–protein interactions, bioapplications, exosomes, viruses

## Abstract

Understanding both the physicochemical and biological interactions of nanoparticles is mandatory for the biomedical application of nanomaterials. By binding proteins, nanoparticles acquire new surface identities in biological fluids, the protein corona. Various studies have revealed the dynamic structure and nano–bio interactions of the protein corona. The binding of proteins not only imparts new surface identities to nanoparticles in biological fluids but also significantly influences their bioactivity, stability, and targeting specificity. Interestingly, recent endeavors have been undertaken to harness the potential of the protein corona instead of evading its presence. Exploitation of this ‘protein–nanoparticle alliance’ has significant potential to change the field of nanomedicine. Here, we present a thorough examination of the latest research on protein corona, encompassing its formation, dynamics, recent developments, and diverse bioapplications. Furthermore, we also aim to explore the interactions at the nano–bio interface, paving the way for innovative strategies to advance the application potential of the protein corona. By addressing challenges and promises in controlling protein corona formation, this review provides insights into the evolving landscape of the ‘protein–nanoparticle alliance’ and highlights emerging.

## 1. Introduction

Nanomedicine, an interdisciplinary area of nanotechnology, is a burgeoning discipline that has garnered significant interest as a comprehensive field of research, offering immense potential for advancements in medical diagnosis and therapeutic interventions [1]. Over several decades, extensive research efforts in drug discovery have resulted in a wide range of powerful therapeutic options for numerous diseases. However, the majority of these drugs have failed to be implemented as standard treatments due to their inadequate effectiveness and potential toxicity [2,3]. One of the primary reasons for this failure can be attributed to the lack of biocompatibility [4].

In the field of biomedical science, nanotechnology has emerged as a highly significant field of research and development in recent times [5,6]. A major goal of nanomedicine is to manage the interaction of nanostructures with biological systems, such as the nano–bio-interface. To achieve this goal, it is significant to figure out the interactions between nanomaterials and their biological environments [1]. The improvement of the caliber of biopharmaceutical agents is a primary goal within the context of the “Quality by Design” paradigm. This principle not only encompasses the targeted clinical use but also its pharmacokinetics, pharmacodynamics, and potential for toxic effects [7,8,9]. Hence, it is important to prioritize the design of drugs that are both efficient and non-toxic [10].

Bioavailability is defined as the degree and speed at which the active drug ingredient or active moiety from the drug product is assimilated and made accessible at the specific location where the drug exerts its effects. The predictive value of the relative bioavailability, which encompasses both the rate and extent of drug absorption, is crucial in determining the potential clinical outcomes [11]. Hydrophobic drug candidates encounter challenges in terms of poor bioavailability, whereas biological-based therapeutics (i.e., proteins and nucleic acids) are prone to rapid degradation. Although altering the chemical composition of a drug can lead to reduced efficacy, nanoparticles (NPs) offer a promising solution. These nano/micro tools not only safeguard the drug but can also facilitate the desired therapeutic outcome [1]. NPs combine advantageous pharmacokinetics with the maintenance of the inherent drug composition [12]. Different kinds of NPs are presently tested in clinical studies [13]. Apart from metallic and inorganic NPs, diverse organic NPs have also demonstrated successful applications in the field of medicine. Two areas contribute most to the economic value of NPs, namely magnetic resonance imaging (MRI) and drug delivery (DD). A large number of NPs have received approval from the FDA or are currently undergoing clinical testing [14,15,16,17].

It is widely accepted in the field that NPs acquire layers of biomolecules, especially proteins, as soon as they come into contact with biological fluids (BFs). Understanding the specific composition of these layers, the so-called protein corona (PC), can be challenging due to its qualitative and quantitative complexity. The development of the PC, which functions as a biological module, involves a highly intricate procedure characterized by specific kinetics and dynamics. Hence, the in vivo performance of NPs is significantly influenced by the formation of a PC, which ultimately determines the biological impact and potential toxicity of these NPs. The composition of PC exhibits notable variations depending on the sources and/or fabrication methods employed for NP systems [14]. For example, the method of gold NP production had an impact on their specific biomolecule binding profiles (albumin, RNA) [18].

The formation of PCs around NPs thus has practical consequences for the use of nanoparticulate tools in living systems (Figure 1). The ‘classical’ stealth coating of NPs, especially by modifications with polymers (polyethylene glycol, PEG, or poly(2-oxazoline), POx), primarily aims at preventing the formation of a PC to increase the half-life of the NPs by escaping the immune response (Figure 1, left). However, as the awareness of the correlation between PC profiles and the in vivo performance of NPs rises, recent endeavors have been undertaken to harness the potential of the PC rather than evade its presence (Figure 1, right) [19]. Several preclinical studies provide evidence that the PC has the potential to facilitate the delivery of drugs to specific organs [20]. Thus, the PC now demands renewed focus as a prospective ally in enhancing the efficacy of NP-mediated drug delivery and turned from a foe to a friend [21]. Understanding and thus exploiting this ‘protein–NP alliance’ has significant potential to change the nanomedical field. It should be noted that both concepts shown in Figure 1 are not sharply separable and can complement each other.

In this review, we aim to compile the latest advances in the understanding of PC formation and structure, as well as to highlight recent innovations in biomedical applications. We also focus on the ‘protein–NP alliance’ showcasing the key points and potential avenues for further advancement of PC exploitation. Key mechanisms, problems, and promises of controlling the protein corona are discussed.

## 2. Who Is Important in the ‘Protein–Nanoparticle Alliance’?

The adsorption process of proteins onto NP surfaces is predominantly affected by the affinity between proteins and NPs, in addition to the interplay of protein–protein interactions transpiring at the NP surface. The structure of PCs can undergo alterations over time. The term ‘hard’ PC describes the internal layer of the PC that interacts with the outer environment over a long period. On the other hand, the ‘soft’ PC represents the outer layer that is not directly connected to the NPs and exchanges with the external medium within seconds or minutes [19]. The adsorption and desorption of proteins within the PC are continuously taking place, primarily regulated by a phenomenon known as the ‘Vroman effect’ [22]. There are two models of the Vroman effect described in the literature (Figure 2). For the ‘new’ approach, a highly complex corona, which can consist of numerous core-shell structures for the early stage of corona formation, is established already within 30 s (Figure 2A, upper panel). In contrast, the ‘old’ model assumes slow evolution of low-complexity corona (Figure 2A, lower panel). In the late stage described by the ‘new’ model, the corona composition is relatively constant ex situ and, due to the Vroman effect, changes predominantly quantitatively rather than qualitatively (Figure 2B, upper panel). The PC undergoes substantial changes over time in the ‘old’ approach, driven by the ‘Vroman effect’, resulting in a highly dynamic system (Figure 2B, lower panel) [23]. Regardless of the specific dynamics of PC formation, it is largely dependent on the physicochemical properties of the NPs, such as surface, size, and shape, as well as the surrounding biological and physicochemical environment. There are several studies that attempt to model the absorption of proteins onto various biomaterials, including NPs, using mathematical models, thus predicting changes in their biological identity [24,25].

The proteins that have been adsorbed are subsequently desorbed and substituted by alternative proteins that exhibit a higher affinity character or binding to NP or PC proteins. Despite altering the composition of the PC, the total quantity of adsorbed protein is relatively stable [19]. The biochemical, physiological, and pharmacological applications of NPs are heavily influenced by their physical properties, including size and surface properties [14] (summarized in Figure 3). Moreover, NP elasticity has also been reported in the literature to be crucial in the physiological fate of NPs, but how this occurs remains largely unknown. Li et al. revealed the mechanisms by which NP flexibility affects the physiological fate of NPs and reported that NP flexibility is an easily tunable parameter in the future rational use of PC [26].

(a)Nanoparticle choice in protein corona

Even in the absence of any pathological conditions, achieving optimal biodistribution and drug delivery is challenging due to various obstacles encountered by NPs. These obstacles include physical barriers, such as shear forces, protein adsorption, and rapid clearance, which collectively restrict the proportion of administered NPs that successfully reach the intended therapeutic site [27]. Disease states frequently lead to modifications in these obstacles, making them even more challenging to overcome through a universal, one-size-fits-all strategy. Moreover, the variations in biological barriers are not consistent among different diseases and can differ significantly from one patient to another. These variations can manifest at systemic, microenvironmental, and cellular levels, posing challenges in their comprehensive identification and characterization. Comprehending the biological obstacles encountered in a general sense as well as on an individual patient basis facilitates the development of highly efficient NP platforms [28].

Proteins and NPs are two important parameters in PC. When making design and selection here, it may also be useful to focus on proteins instead of NP selection, which is a very comprehensive and different issue in itself. While many parameters are involved in the PC interaction, it is extremely interesting to know that proteins will have different net charges at different medium pH. Instead of selecting a large number of NPs, it may be more beneficial to design PCs by focusing on proteins with the known net charge at a specific pH or isoelectric pH points. Besides active electrostatic interactions, other secondary interactions are known to exist in PC. Of course, interacting with NPs by adjusting the net charge of the protein will increase the probability of interaction, and the desired interactions will be achieved thanks to NP–protein engineering. However, without ignoring the NP side, we have touched on some important parameters of some NPs, such as surface, size, and shape, in the following sections.

(b)Effects of surface, size, and shape of NPs

Surface area plays a significant duty in various interactions. Typically, NPs possess a substantial surface area, which can prove beneficial when it comes to nonspecific interactions with serum proteins [14]. Recently, there is evidence that platelet aggregation is not related to NP size but depends on the surface area of NPs [29]. Furthermore, the substantial proportion of the overall accessible surface area of NPs and protein concentration cannot be solely attributed to the robust adsorption and binding of plasma proteins. An important example of this is the finding that silica NPs bind to more plasma proteins at lower concentrations of plasma proteins than polystyrene NPs of the same size. However, it was discovered that their ability to bind was reduced in the presence of elevated levels of plasma proteins in the surrounding environment. The polystyrene nanoparticles displayed a contrasting pattern in their behavior [30].

The characteristics and potential interactions of nanomedicines with proteins in BFs are significantly influenced by both their size and shape (Figure 3B). Specifically, these two parameters play a vital role in determining the behavior and efficacy of nanomedicines within complex media, such as proteins in BFs. The impact of NP size extends beyond the quantity of proteins bound to their surfaces, also influencing the categories of proteins involved. Shape profiles also have the potential to alter the mass/surface ratio (curvature) of nanomedicines. Typically, the size of nanodrugs governs their mass/surface area, and surface area is determined by their size [31,32]. Larger particles exhibit a reduced surface curvature a larger surface area, and provide more surface interaction opportunities for each protein [33,34,35]. This allows for broader coverage of proteins. Conversely, smaller NPs display an enhanced surface curvature, leading to a reduction in conformational alterations, advancement of constituents, and higher levels of adsorbed proteins. Additionally, this phenomenon leads to an expansion of the corona width [1,31,36,37,38,39]. To elaborate, smaller particles exhibit higher surface curvature, smaller surface area, and limited surface interaction with proteins. This results in a lesser degree of protein coverage. Larger particles have lower surface curvature, larger surface area, and provide large surface interaction for proteins, thus facilitating protein coverage to a greater extent [1].

(c)The biological environment

Nanomaterials possessing hydrophobic and/or charged surfaces frequently exhibit a higher propensity for protein adsorption and subsequent denaturation compared to neutral and hydrophilic counterparts. The protein structure changes depending on the charge and is positive > negative > neutral [1,40]. NPs possessing a significantly high positive charge and an isoelectric point lower than 5.5 demonstrate a swift and strong interaction with proteins, whereas NPs with a high negative charge predominantly interact with substances having an isoelectric point greater than 5.5 [41]. Hydrophobic surfaces have the ability to induce protein denaturation or structural alterations by compelling the unveiling of their hydrophobic regions. Consequently, this also aids in augmenting the quantity of adsorbed proteins [42,43]. In addition, the resultant protein adsorption patterns undergo a qualitative transformation, enhancing the affinities of biomolecules and inducing protein conformational changes. Additionally, they exhibit a faster opsonization rate compared to hydrophilic NPs [44]. The roughness of the surface significantly reduces the repulsive interactions, leading to a notable impact on the quantity of proteins while leaving their identity unaffected [1,45]. As a result, while the amount of protein accumulated/adsorbed on the surface of the nanostructures changes, it does not have any effect on the structure and character of the nanostructures.

In biomedicine, NPs encounter various “complex” surroundings upon administration in vitro/in vivo. Cell culture medium (CCM) is the prevailing environment that NPs commonly encounter under in vitro conditions. This medium typically comprises fetal calf or bovine serum, which is essential for promoting optimal cell growth. The evaluation of the performance and behavior of particle systems in simulated physiological solutions may be required, depending on the specific biomedical application. Moreover, it is crucial to comprehend the potential scenarios that NPs are able to confront in whole blood or plasma when they enter the human circulatory system. This understanding plays a pivotal role in guiding the logical development of particle systems. The human body contains a multitude of dynamic mechanisms that can effectively stabilize, solubilize, and modify particles through biomacromolecules, changes in ionic strength, alterations in pH, and active biological processes [46].

The surrounding biological environment plays an important role in determining the potential biological impacts of nanostructures utilized in the biomedical field, in addition to their physicochemical properties [47]. In physiological fluids, NPs are very likely to enter into various colloidal interactions with various constituents (such as salts, sugars, and proteins). This “nano-bio interface” involves dynamic physicochemical interactions, kinetic factors, and thermodynamic exchanges between the surfaces of nanostructures and biological components. The in vivo stability of nanostructures can be significantly impacted by these interactions, leading to devastating consequences. Basic physical and chemical investigations are often carried out in controlled settings to preserve the entirety of the design of particle and conjugation. Nevertheless, the ultimate determination of the biocompatibility and biodistribution of these particles relies on the alterations that take place within the particle upon exposure to physiological fluids. Upon taking the body, nanoparticles (NPs) have the potential to experience a range of transformations, including protein adsorption, dissolution, agglomeration/aggregation, structural deformities, and redox reactions. These alterations play a crucial role in influencing the absorption, availability, movement, and ultimate destiny of NPs, which in turn dictate their effectiveness in therapy, diagnostic accuracy, or potential toxicity [46]. After entering the physiological medium, which includes blood, interstitial fluid, and intracellular environment, various forces mainly exist in the interaction of NPs with biomolecules, including van der Waals forces, hydrogen bonds, electrostatic, and hydrophilic/hydrophobic interactions. These forces are attributed to the high surface energy and distinctive surface chemistry of the NPs. As a result, it subsequently gives rise to the occurring of a biomolecular corona structure [47].

Comprehending the characteristics of both soft and hard coronas is of utmost importance in comprehending the stability, functionality, and interactions of nanoparticles (NPs) with biological systems. From a kinetic standpoint, the interactions between proteins and NPs in biological fluids are regulated by non-covalent forces, including electrostatic forces, hydrophobic forces, hydrogen bonding, and π-π stacking. Proteins competitively bind to the surfaces of NPs, resulting in the formation of transient NP–protein complexes that consist of both soft and hard corona proteins under thermodynamically favorable conditions. However, due to the rapid dissociation rate of soft corona proteins, the current understanding of the biological composition of the protein corona is typically limited to hard corona proteins [48].

(d)The physicochemical environment

It is accepted that pH and temperature conditions during in vitro incubation strongly affect the affinity of protein for NPs (Figure 3C). Lately, Gorshkov et al. put forward the notion that PCs are composed of a diverse range of proteins, including those that are resistant as well as those sensitive to changes in temperature or pH. The effect of temperature on protein diffusion and their affinity for NPs can be observed even in the physiological range of approximately 37 to 40 °C/41 °C [49]. Oberländer et al. demonstrated that surface charge and surfactant composition could be influenced under conditions of constant temperature and concentration. Furthermore, they observed that the corona structure formed at low temperatures (4 °C) differs from that formed at physiological temperatures (37 °C). Their findings also indicated that the uptake of nanoparticles by model cancer and endothelial cells decreased with increasing temperature or plasma concentration, irrespective of nanoparticle formulation [50].

On the other hand, alterations in the pH effect may cause changes in the protein structure on the NP surface. The pH levels in various biological compartments differ, ranging from acidic to neutral or slightly basic. The stability of PC is influenced by pH due to the presence of salt bridges and hydrogen bonds, which play a crucial role in the folding of PC protein [51]. The composition of PCs can vary significantly when NPs are exposed to BFs in vivo or in vitro [52].

It has been proven that the protein composition in the PC is significantly influenced by static conditions such as flow rate and dynamic conditions. The PC exhibits less homogeneity during dynamic circumstances, allowing certain NPs to remain uncoated and available for interaction with cells [53].

(e)Importance of protein corona composition

The literature lacks a unanimous agreement regarding the benefits and drawbacks of protein adsorption [54]. The PC’s composition is intricate, diverse, and heavily influenced by the specific biological surroundings it interacts with upon contact with NPs, suggesting the potential for exposure-induced memory. Specific elements within the PC, like Opsonins (IgG, complement, among others), could enhance the rapid uptake of coated nanoparticles by the reticuloendothelial system [55,56]. Moreover, proteins known as dysopsonins, like albumin, have the ability to inhibit complement activation when bound to particles. This results in an extended circulation period and decreased toxicity [54]. Despite the fact that some proteins are unique to particular nanoparticles and the protein corona’s composition is influenced by nanoparticle characteristics, albumin, IgG, fibrinogen, and apolipoprotein are consistently found in the majority of PC studies [57]. The PC’s composition is contingent upon the concentration and kinetic characteristics of the plasma proteins in a time frame. Consequently, comprehending the dynamics of corona protein exchange becomes crucial as it dictates the PC’s significance within the broader biological profile of nanoparticles [58]. It is important to recognize that the modification of the secondary structure of proteins, which affects the overall bioreactivity of NPs, is influenced by various factors. These factors include the surface area and flexibility of the NPs, as well as the chemical properties of the absorbed proteins. NPs with curved surfaces, as opposed to planar surfaces, offer greater flexibility and a larger surface area for the adsorption of protein molecules [59]. Furthermore, the NPs’ curved surface could potentially affect the protein’s secondary structure, leading to irreversible alterations in certain instances [60]. To give an example, while the effect of gold NPs on the conformational changes and concentration-dependent behavior of BSA was demonstrated [61], in contrast to this situation, it was reported that no changes in the conformational changes of BSA could be detected when carbon C60 fullerenes were absorbed on the surface of NPs [62]. Furthermore, regarding this issue, it has been noted that conformational changes in the secondary structure of transferrin are irreversible when exposed to ultrasmall superparamagnetic iron oxide NPs [63].

The findings indicate that apart from the process of protein association with NPs, there exists another crucial aspect that remains inadequately elucidated, namely the interaction between proteins and the particle surface. While the significance of hydrophobic/hydrophilic and electrostatic interactions is acknowledged, a comprehensive description of these interactions is currently lacking. A thorough comprehension of these interactions holds the potential to enhance our understanding of the factors influencing the kinetics of protein binding. Consequently, there is a necessity to enhance analytical procedures and techniques to comprehensively elucidate the intricate processes involved in the interaction between proteins and nanoparticles [54].

## 3. How Can We Exploit This Alliance?

The correlation between the physicochemical properties of NPs and their biological interactions remains an unresolved enigma within the realm of nano–bio interactions. The surface feature of the designed NPs primarily determines the bioactivity in targeted DD. Even a slight alteration in physicochemical properties has the potential to alter the biological effect of NPs and, as a result, lead to a poor in vitro–in vivo correlation [64]. Instead of solely focusing on ‘fighting against’ PC formation, we should leverage the ‘protein–NP alliance’ for the sake of improved nanomedical applications. In the following section, we discuss how biomedical researchers can exploit this alliance, especially in consideration of the PC’s effect on cellular uptake, cytotoxicity, DD and targeting, immune response, as well as its reflection of (patho)biological conditions.

(a)Fine-tuning cellular uptake

One of the key factors for the development of efficient DD systems to treat various diseases is the internalization of NPs by cells. The presence of a dynamic PC in cells can have a great impact on the sequence of events during NP–cell contact. This includes processes like cellular internalization and pathway activation, which may be notably affected [64].

Wu et al. [65] explored the cellular internalization and toxic impact of MXene Ti_3_C_2_T_x_ and polyethylene glycol (PEG)-modified Ti_3_C_2_T_x_ in THP-1 cells, mediated by the presence of a human serum protein corona (Figure 4). They presented the possibility that the PEGylation process could change the interactions between Ti_3_C_2_T_x_ and serum proteins and thus cause a major transformation in the PC fingerprint. They found that PEGylation can alter the interaction between Ti_3_C_2_T_x_ and serum proteins, and PEGylation and PC formation increased MXene cellular uptake. After PC formation, Ti_3_C_2_T_x_ and PEGylated Ti3C2Tx were observed to accumulate mostly in lysosomal regions within THP-1 cells (Figure 4, right). Furthermore, clathrin-assisted endocytosis emerged as the internalization of Ti_3_C_2_T_x_ by THP-1 cells, which is the main process through which this material is taken up by the cells. They reported that no significant effect on cell viability was observed. In addition, Ti_3_C_2_T_x_ has a dual effect as both a stimulator and scavenger of ROS in THP-1 cells, which are affected by PEGylation and PC formation [65].

The cellular destiny of the nanocarriers is influenced by alterations in the composition of the PC, which can be contingent upon certain parameters. Oberländer et al. [50] reported a study examining the impacts of three crucial factors, such as charge of surface, temperature, and concentration of plasma, on the development and composition of the PC surrounding polystyrene NPs. It was noted that the composition of the corona varies between low-temperature values (4 °C) and physiological temperature values (37 °C). Interestingly, at lower plasma concentrations (up to 25%), the composition of proteins in the corona is more variable compared to higher concentrations. Ultimately, their findings indicate that irrespective of the formulation of the NPs, the uptake of these particles by model cancer and endothelial cells decreases when they are pre-coated at higher temperatures or plasma concentrations [50]. Although these results may be limited to the analyzed NPs, a potential influence of coating temperature and pre-incubation should be considered when aiming to deliberately alter the cellular uptake of nanosystems.

Sodium citrate-stabilized gold NPs (AuNPs) experience a loss of stability upon dispersion in cell culture media (CCMs). This phenomenon has the potential to facilitate the aggregation of particles and their subsequent sedimentation. Under the appropriate circumstances, the interaction between AuNPs and dispersed proteins may result in the formation of a stabilizing PC. CCMs are aqueous ionic solutions containing growth substances, often combined with various additives such as antioxidants, dyes, and antibiotics. Barbero et al. conducted a study to examine the influence of additives, namely phenol red, penicillin–streptomycin, l-glutamine, and β-mercaptoethanol, on the development of the PC in cell culture media [66]. PCs with comparable characteristics were acquired, with the exception of the inclusion of antibiotics. Under such circumstances, the PC’s formation experienced a delay, and certain investigations demonstrated alterations in both its density and composition. As a result of these alterations, a markedly increased cellular uptake of AuNPs was observed combined with the formation of NP aggregates. Investigations of the internalization mechanism responsible for AuNP uptake suggested that this process was not significantly influenced by either clathrin receptors or lipid rafts. These findings indicate that under cell culture conditions, the primary factor influencing cellular uptake is NP aggregation [66].

In addition to cell culture media and blood, various other biological fluids are relevant for protein corona formation and, thus, for potential biomedical applications. Von Mentzer et al. aimed to evaluate the impact of robust PC structures derived from synovial fluids of patients with osteoarthritis and rheumatoid arthritis, as well as from widely utilized fetal calf serum (FCS), on the internalization of NPs into tissues and cells. To achieve this, they created a panel of NPs with varying degrees of PEGylation, which were then incubated with synovial fluid from osteoarthritis or rheumatoid arthritis patients or with FCS. The uptake of the NP-PC structures had diverse effects on porcine articular cartilage explants, chondrocytes, monocyte cell lines, and primary patient fibroblast-like synoviocytes (FLS) cells. The uptake of NPs into cartilage tissue and their internalization in chondrocytes and monocytes were significantly influenced by the resulting biocoronas. This experiment highlights the impact of PCs derived from synovial fluids of osteoarthritis or rheumatoid arthritis patients on NP uptake into cartilage. Furthermore, it underscores the importance of considering the biological microenvironment to successfully translate drug delivery tools into clinical practice [67].

(b)Decreasing Cytotoxicity

Different NPs exhibit varying degrees of cytotoxicity depending on their material and physicochemical factors discussed above. For example, while some metallic NPs like silver and copper can induce cytotoxicity and oxidative stress, others like silica NPs may have lower toxicity profiles but can still elicit inflammatory responses in certain conditions [68]. Therefore, the control of cytotoxicity is an important factor in the use of nanoformulated drugs.

Food-borne carbon dots, naturally occurring during the thermal processing of food, are of significant importance due to their impact on human health. Upon oral administration, it is expected that these carbon dots will interact with blood proteins, resulting in the spontaneous formation of PCs. In a pioneering study, Cui et al. investigated the interaction between carbon dots derived from roasted mackerel and three major blood proteins: albumin, gamma globulin, and fibrinogen [69]. Their main objective was to evaluate the influence of these carbon dots on the physiological effects such as cytotoxicity and metabolic response. The results revealed that food-borne carbon dots readily bind to all of the three analyzed blood proteins, creating multiple interaction forces in the process. Furthermore, three PCs demonstrated the ability to mitigate the adverse effects of food-borne carbon dots on cell viability, oxidative stress, and mitochondrial membrane potential. This study holds significant importance in assessing the toxicity properties of food-borne carbon dots and their specific PC.

The potential release of rare earth elements (REE) into the environment and their subsequent ingestion by humans has raised concerns due to their widespread use. Therefore, it is crucial to assess the cytotoxicity of these elements. Feng et al. conducted a study investigating the effects of three typical REE ions (La, Gd, and Yb) as well as nanometer- and micrometer-sized oxides on red blood cells, which are potential targets when introduced into the bloodstream [70]. They examined hemolysis induced by REE at concentrations ranging from 50 to 2000 μmol/L to simulate cytotoxicity under medical or occupational exposure scenarios. The study revealed that the occurrence of hemolysis due to REE exposure is significantly dependent on their concentration. Furthermore, the observed cytotoxic effects followed a specific order, with La^3+^ having the highest impact, followed by Gd^3+,^ and then Yb^3+^. Interestingly, while the cytotoxicity of REE ions exceeded that of rare earth element oxides (REOs), nanometer-sized REOs induced more hemolysis than micrometer-sized ones. Investigations into the mechanism showed that REE caused cell membrane rupture through reactive oxygen species (ROS)-associated chemical oxidation, as evidenced by the generation of ROS, ROS quenching assays, and detection of lipid peroxidation. Additionally, it was suggested that the formation of a protein corona on REE increases steric repulsions between these elements and cell membranes, thereby reducing their cytotoxicity [70].

Silver nanoparticles (AgNPs) exhibit remarkable versatility, finding applications across various biomedical fields. Like other NPs upon interaction with the biological milieu, AgNPs undergo protein adsorption, forming biomolecular coronas. Batista et al. conducted a study examining the impact of protein adsorption on the surface of polymer-stabilized AgNPs [71]. Their investigation revealed that the predominant component of PCs was bovine serum albumin, the most abundant protein in the model biological environment. In addition, the researchers demonstrated a notable decrease in the toxic effects of silver colloids once enveloped with biomolecular coronas compared to their original, uncoated counterparts [71].

Da Silva et al. [72] synthesized titanium dioxide NP–multi-walled carbon nanotubes (TiO_2_—MWCNT) and conducted a pioneering study investigating their cytotoxic effects, PC formations, and cellular uptake on fibroblasts derived from gonadal rainbow trout tissue (RGT-2) (Figure 5). Remarkably, this nanohybrid exhibited no toxicity towards RTG-2 cells even at high concentrations and 24 h of exposure [72].

Collectively, recent studies have corroborated that diverse types of NPs interact with proteins present in different biological environments, thereby acquiring distinct PCs. Furthermore, the majority of these studies have demonstrated that PCs have the capability to mitigate NP-induced cytotoxicity.

(c)Improving Drug Delivery and Targeting

Nanomaterials exhibit tremendous potential as drug carriers or active agents for DD and disease treatment [73,74,75,76,77]. Their remarkable characteristics, including small size, versatile functionalization, targeting ability, and controlled release capability, render them particularly valuable in this domain [78,79,80,81].

Liposomes, recognized as a promising category of antibiotic delivery systems, have demonstrated efficacy in managing severe bacterial infections that present life-threatening risks. However, the inevitable formation of a PC on the liposome surface may significantly influence their performance in vivo. The potential to advance the development of antibacterial liposomal drugs is substantial, as it can significantly improve our understanding of how the presence of a PC affects liposome behavior. In a study conducted by Shao et al., they elucidated the role of the PC in facilitating liposome–bacteria interactions [82]. Negatively charged proteins increased liposomal binding to bacteria by attenuating the electrostatic attraction through their adsorption to the cationic liposome. Moreover, the strong binding affinity of anionic liposomes composed of phosphatidylglycerol (DSPG sLip) to both planktonic bacteria and biofilms was facilitated by the cumulative complement accumulation on the DSPG sLip. This approach has been employed to enhance bacterial-targeted DD. In mouse models of both *Staphylococcus aureus*-associated osteomyelitis and pneumonia, DSPG sLip demonstrated promise as an antibiotic nanocarrier for the management of resistant *S. aureus* (MRSA) infection. This further underscores the utility of PC modulation through lipid composition in liposomal antibiotic delivery for bacterial infection treatment [82].

The arrangement of PCs on the surfaces of NPs plays a crucial role in regulating physiological interactions, including cellular association and targeting capability. In another study, repeated administrations of α-mangostin (αM)-loaded poly(ethylene glycol)-poly(l-lactide) (PEG-PLA) NPs (NP-αM) effectively enhanced the expression of low-density lipoprotein receptor (LDLR) in microglia. This, in turn, leads to improved clearance of amyloid beta (Aβ). However, the mechanisms by which NPs traverse the blood–brain barrier and reach microglia remain elusive. Tang et al. investigated the brain delivery characteristics of PEG-PLA NPs in various conditions [83]. Their findings revealed that NPs exhibited enhanced efficiency in brain transport and uptake by microglia after αM loading and multiple administrations. To elucidate the underlying process, a proteomic study was conducted to depict the structure of the PC in different scenarios. The results obtained showed that both drug loading and multiple doses administration affect the composition of the PC, thereby affecting the uptake of NPs in endothelial cells (b.End3) and B-lymphocytes (BC-3). Another significant finding of the study is that the PC is enriched with complementary proteins, immunoglobulins, RAB5A, and CD36, which are related to the NP uptake process [83].

PEGylated lipid-based nanocarriers have garnered widespread recognition for their remarkable capacity to enhance DD efficacy while simultaneously reducing toxicity. However, it is crucial to note that the presence of a PC on the nanosurface can potentially alter their biological characteristics, leading to unpredictable outcomes in vivo. Specifically, the attachment of pre-existing anti-PEG antibodies may accelerate the removal of PEGylated nanocarriers from the bloodstream via complement activation and opsonization. This process ultimately diminishes their effectiveness and may provoke unforeseen negative immune responses. Achieving efficient and safe drug delivery necessitates precise adjustment of the structure and role of the PC. Chen et al. uncovered that lipid nanodiscs with a discoid shape exhibited a distinct PC pattern, resulting in significantly different biological outcomes compared to spherical liposomes. The discoid structure played a pivotal role in minimizing complement binding and activation, thus enabling evasion of the detrimental phenomenon of accelerated blood clearance. This phenomenon profoundly influences the in vivo efficacy of PEGylated nanocarriers. Moreover, the discoid surface exhibited a preference for the adsorption of apolipoproteins, which retain their functional properties. This unique characteristic of nanodiscs endows them with a specific ability to target the brain. Overall, the research presented in this study provides novel insights into how the morphology of lipid-based nanocarriers could be exploited to manipulate the PC, ultimately enhancing targeted DD [84].

In summary, understanding the interplay between proteins and NPs is crucial for the rational advancement of targeted DD systems, as the distribution and behavior of NPs in vivo can be significantly influenced by the formation of a PC. By comprehending these protein–NP interactions, researchers can design platforms that efficiently deliver drugs to specific targets. The application of PEG on the surface of NPs has been found to be effective in reducing PC formation and enhancing the efficiency of DD. However, these/other techniques are not sufficient to completely eliminate PC. However, these techniques alone may not completely eliminate PC formation. Therefore, strategies aimed at enhancing the efficacy of PCs for drug transportation may yield greater efficacy in the future. Prior to injection, the desired PC can be engineered around NPs, or NPs can be functionalized to induce the formation of a PC composed of de-opsonin, such as albumin, transferrin, or ApoE in vivo. This type of PC has the ability to inhibit opsonization, thereby preventing the phagocytosis of NPs. In general, the choice of a suitable PC depends on the intended application and function, either prolonging the half-life of the NPs in a biological system by increasing specific targeting or cellular uptake. Here, more detailed studies are mandatory to reveal general principles of targeted PC design [19].

(d)Advancements in cancer diagnosis

The burgeoning focus on understanding the functional roles of PC structures in tumor diagnosis has gained momentum alongside the widespread growth of PC research. Across various tumor types, unique proteins have the potential to be expressed, leading to altered concentrations of these proteins in the plasma [85,86]. Consequently, when NPs are incubated in these plasma samples in vivo, proteins are expected to adhere to and accumulate on them, thereby influencing the composition of their PC. While detecting diagnostic biomarkers and assessing their therapeutic efficacy can be challenging in oncological studies [87], harnessing the PC of NPs for tumor detection presents a promising, straightforward approach.

Zhang et al. introduced novel insufficient polyhedral oligomeric silsesquioxane (POSS) polymer-caged AuNPs (PP-AuNPs) for the detection of metallothioneins, pivotal biomarkers in various tumors [88]. The POSS polymer network’s exceptional sensitivity to H_2_S as a reducing agent was leveraged in this approach. Upon incubation in biological samples, the researchers extracted the PP-AuNPs bound with PC to catalyze the reduction of 4-nitrophenol. If metallothioneins were present in PC, the reduction process would be impeded, or the conversion of 4-nitrophenol to 4-aminophenol would occur rapidly, resulting in a color change due to the lack of metallothioneins.

In another study by Papi et al. [89], liposomes combined with 1,2-dioleoyl-sn-glycero-3-phospho-(10-rac-glycerol) (DOPG) were utilized for the early detection of central nervous system tumors. Specific proteins such as Integrin beta-2 and lactotransferrin enabled the sensitive differentiation of PC profiles between meningeal tumor patients and healthy individuals. The identification of these proteins in PCs holds promise for their potential use as biomarkers for brain tumors.

Blood tests based on the PC come with inherent limitations that warrant acknowledgment. The protein component of PCs is not exclusively influenced by tumors; it can also be impacted by various other (patho)physiological conditions. Consequently, these fluctuations have the potential to alter the dynamics of protein interaction associated with tumors, compromising the precision of detection outcomes [90]. NPs designed for tumor diagnosis relying on PCs have been developed using diverse standards. However, a notable limitation arises as protein databases often lack comprehensive information on various tumor types, impeding their potential for clinical implementation. Furthermore, there is a possibility of significant delays between test initiation and the receipt of results for PC-based detection, lacking the necessary promptness required in clinical settings. Thus, efforts should be made to minimize this delay and align it with clinical requirements. In conclusion, by integrating multidisciplinary approaches, the utilization of PCs on nanostructures as innovative biomarkers for rapid and seamless tumor detection holds promise for extensive adoption, potentially yielding precise outcomes in the future [91].

(e)Modulating immune cells

Despite numerous studies on PC dynamics, the precise impact of distinct PC compositions on cells, particularly immune cells, is not fully understood. Macrophages play a crucial role in the immune system, exhibiting the ability to polarize into either a pro-inflammatory (M1) or anti-inflammatory (M2) phenotype [92]. In a study by Yang et al., gold NPs (AuNPs) were incubated with human plasma for varying durations to form time-dependent PCs [92]. Thus, an investigation was conducted to analyze the effect of these time-dependent AuNP-coronas on macrophage polarization (Figure 6). Results from morphology assessments, biomarker analyses, and cytokine secretion assays demonstrated that intact AuNPs and 4 h-AuNP-coronas promoted macrophage polarization towards the M2 phenotype. Conversely, co-incubation of 12 h-AuNP-coronas with macrophages induced an M1 phenotype. Additionally, proteomic investigations revealed continuous changes in PC compositions following AuNP interaction with plasma. Notably, prolonging the incubation period to 12 h led to a substantial increase in immune proteins within the PC, thereby modulating the polarization of different macrophage subsets [92].

## 4. Does This Alliance Also Extend to Other Nano-Sized Structures?

In addition to naturally occurring and artificially synthesized NPs, various structures within the nano-scale range interact with BFs and consequently acquire specific PCs. Among these structures are small extracellular vesicles known as exosomes, typically ranging in size from 30 to 300 nm [93]. Upon thorough investigation of exosomes, their pivotal roles in intercellular communication and cellular immune response become evident [94]. Often referred to as cellular “waste bags”, exosomes are released to eliminate membrane proteins deemed non-essential by cells [95]. They serve as natural membrane vesicles transporting protein, RNA, DNA, and lipids from one cell to another, thereby regulating the biological functions of recipient cells [96,97,98]. Compared to synthetic nanomaterials, exosomes, as natural counterparts, offer significant and distinctive advantages. Due to their natural function as transport vehicles, exosomes are optimized in structure and composition for specific intercellular transport and cargo delivery and, thus, are not prone to be eliminated by the immune response. For example, exosomes can traverse the blood–brain barrier, allowing DD to the central nervous system [99,100,101].

Given that almost all cell types release exosomes, they are highly abundant in BFs, where they bind biomolecules, such as proteins. However, the detailed mechanism underlying the formation of PCs on exosomes remains incompletely understood. Existing literature has demonstrated PC formation around exosomes in aqueous phases, primarily through electrostatic interactions and protein aggregation [102]. However, the understanding of the factors influencing PC formation in biofluids remains both theoretically and experimentally underdeveloped [102,103]. Previous studies have yielded conflicting results regarding the presence of albumin on the surface of exosomes [103]. Albumin interacts with various factors, including fibronectin, the complement system, and prothrombin, to bind to the surface of exosomes. This interaction leads to the formation of dense exosomes with a distinct PC [104]. Interestingly, pathological conditions often lead to a notable decrease in the albumin/globulin ratio. For instance, patients with chronic obstructive pulmonary disease exhibit a lower albumin/globulin ratio compared to healthy individuals attributed to chronic inflammation and narrowed small airways [105].

During the formation of PCs, biological responses are primarily mediated by associated proteins rather than NPs, maintaining a delicate balance in vivo. Furthermore, the surface composition of NPs is subject to unpredictable alterations, leading to unforeseeable immunological responses [106,107]. Opsonins, integral components of the PC, readily bind to NP surfaces, categorizing them as “foreign entities” and hastening their removal from the bloodstream [108]. Although PEG coating has proved effective in reducing PC, the elimination of PEGylated NPs accelerates, especially with repeated administrations, potentially due to the development of α-PEG antibodies [109,110]. In contrast, exosomes are not recognized as foreign by the immune system, thus evading eradication [93]. Unanswered questions persist regarding the potential impact of the PC on the bioactivity of exosomes, especially in physiological conditions. Moreover, previous experiments have shown that chemically modified exosomes tend to exhibit enhanced delivery efficiency [102,111]. Prospectively, detailed characterization of specific PCs formed around different species of exosomes, such as those derived from immune cells or tumors, is mandatory. This endeavor would not only deepen our understanding of the functionality of these natural nanocarriers but also offer new ideas for the potential therapeutic use of artificial exosomes or liposomes.

In addition to exosomes, viruses serve as naturally occurring nano-sized structures utilized for biomolecule transport. The inherent protein coating of viruses facilitates the formation of a natural PC. Importantly, artificial NPs acquire a PC layer upon exposure to BFs, significantly influencing their bioactivity. Ezzat et al. [112] highlighted the similarities between viruses, as nano-sized obligate intracellular parasites, and artificial NPs in extracellular environments. They observed that respiratory syncytial virus (RSV) and herpes simplex virus type 1 (HSV-1) exhibited distinct behaviors and PC formations in various BFs. Furthermore, they found that corona pre-coating had differential effects on viral infectivity and immune cell activation. Their findings underscored the importance of the viral PC as a crucial structural layer for viral–host interactions, revealing mechanistic similarities between viral and amyloid pathologies.

The intravenous administration of oncolytic adenoviruses holds great promise as a tumor treatment method [113]. However, the efficacy of these viruses is often compromised by the immune system’s efficient elimination of them. Numerous investigations have focused on extending the circulation of intravenously administered oncolytic adenoviruses by preventing their binding to neutralizing antibodies and complement factors in the blood. Unfortunately, these efforts have yielded unsatisfactory results [113]. In contrast, Huang et al. [113] discovered that hindering the formation of virus-PCs rather than preventing the binding of neutralizing antibodies or complement factors to oncolytic adenoviruses is key to improving their circulation. After identifying the essential protein components of the virus-PC, they employed a virus-PC replacement approach, creating an artificial virus-PC on oncolytic adenoviruses to block their interaction with key virus-PC components in plasma. Serum albumin (SA), fibrinogen (FB), and immunoglobulin (IM), the most common and abundant proteins in the nanoparticle PC, were found to be significantly negatively correlated with most of the major virus protein corona components identified in this study. This method significantly extended the circulation time of the adenoviruses and enhanced their distribution in tumors. As a result, the compound exhibited significantly improved antitumor effectiveness in both primary and metastatic tumor models. This discovery offers a new perspective on the intravenous administration of oncolytic adenoviruses, shifting the focus of future research from inhibiting the binding of oncolytic adenoviruses with neutralization antibodies to fine-tuning their interaction with crucial constituents of the virus-PC in plasma.

## 5. What Is Next?

When nano-scale objects encounter BFs, they undergo a transformation, shedding their original identity and adopting a new biological identity. This transformation, known as PC formation, profoundly alters various physicochemical characteristics of NPs, including their surface charge, size, and aggregation state. Consequently, these alterations exert a significant influence on the biological fate of NPs, encompassing their biodistribution, pharmacokinetics, and therapeutic efficacy. It is widely accepted that even slight variations in the composition of protein-containing fluids, such as plasma and serum, can substantially modify the structure of the PC that develops on the surface of NPs. The ‘protein–nanoparticle alliance’, arising from this specific interaction between proteins and NPs proves highly beneficial in numerous biomedical contexts (Figure 7). This term is coined to emphasize that while proteins and nanostructures may have limited utility independently in certain applications, their combined use offers a versatile approach across a diverse array of applications. For example, recent studies have unveiled the existence of disease-specific PCs, wherein the pathological conditions of patients manifest in distinct PC compositions. This groundbreaking discovery has paved the way for the concept of personalized PCs. Emerging research has underscored the potential of generating personalized PCs as a promising approach in the realm of personalized medicine, showcasing its profound significance in clinical applications [114]. Hence, comprehending the interplay between nanomaterials and biological fluids is fundamental for their utilization. The utilization of proteomics, bioinformatics, and nanotechnology in conjunction can aid in identifying the components of the PC that are created when metal NPs are exposed to biological fluids. This approach holds the potential to offer valuable insights into disease progression and facilitate the exploration of novel therapeutic targets [114].

The utilization of PC in NPs for tumor detection could offer a straightforward and intelligent approach. Compared to traditional biomarker strategies, relying on PC on NPs has the potential to stand out as a strategy that does not require laborious steps [115]. Moreover, PCs can elucidate alterations in protein concentration caused by tumor growth and invasion, enabling early and minimal-invasive diagnosis [116].

Recent advancements in the identification and separation techniques of extracellular vesicles (EVs) have opened up new avenues for innovative therapeutic strategies. Among the spectrum of EVs, exosomes stand out for their capacity to transport diverse signaling biomolecules and exhibit numerous advantageous characteristics compared to therapies based on whole cells. Given the diverse adsorption capabilities of numerous molecules onto the surface of exosomes and the inherent variations in serum proteins among individuals, substantial variations in corona thickness are expected. These variations will profoundly influence the kinetics, biodistribution, docking, and cellular internalization of exosomes. Considering that the primary constituents of the PC consist of typical serum proteins, it is reasonable to speculate that the addition of PC to allogeneic immunogens could potentially shield the administered exosomes from the immune system. However, it is crucial to consider the activity of additional pro-inflammatory cytokines. In contrast to synthetic NPs, the complexity of the exosomal membrane, along with its various ligands, contributes to a high heterogeneity of the PC and other factors bound to the exosome surface. In summary, the development of a PC surrounding exosomes has the potential to alter their physicochemical characteristics and potentially impact their targeting ability. The composition and capacity of the PC may vary significantly due to genetic variations among individuals, variances in exosomes from different donors, and the protein profile in the recipient’s serum [102].

In conclusion, especially exploiting the personalized nature of PC composition is a promising strategy for personalized medicine, offering tailored approaches for disease diagnosis and treatment. However, challenges remain in fully understanding and harnessing the complexities of the protein–nanoparticle alliance. Future research efforts will undoubtedly focus on unraveling these complexities and leveraging this alliance to advance biomedical science and improve patient outcomes.

## Figures and Tables

**Figure 1 nanomaterials-14-00823-f001:**
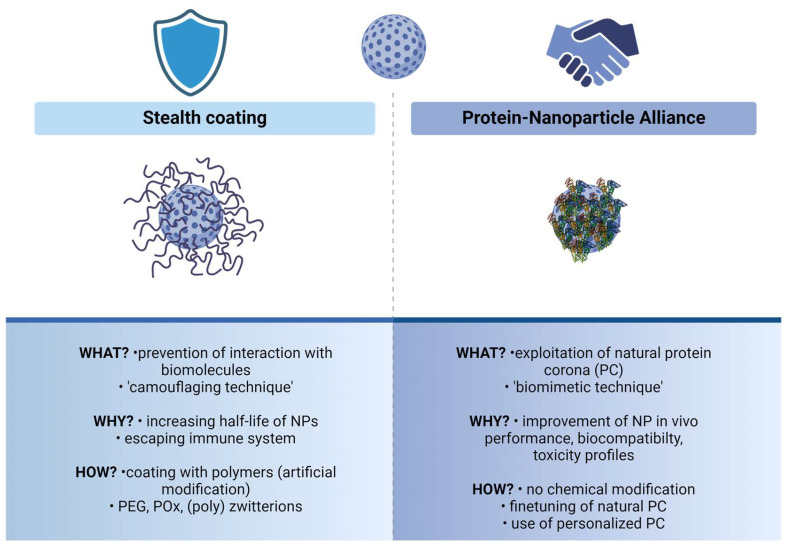
Two complementary concepts of understanding and working with the protein corona formed around nanoparticles (NPs). The ‘classical’ stealth coating of NPs primarily represents a ‘camouflaging technique’ seeking to ensure evasion of the immune response. Additionally, understanding this system as a beneficial ‘protein–NP alliance’, opens up new possibilities for utilizing and fine-tuning the naturally formed protein corona (PC) of NPs. PEG, polyethylene glycol; POx, poly(2-oxazoline). Created with BioRender.com.

**Figure 2 nanomaterials-14-00823-f002:**
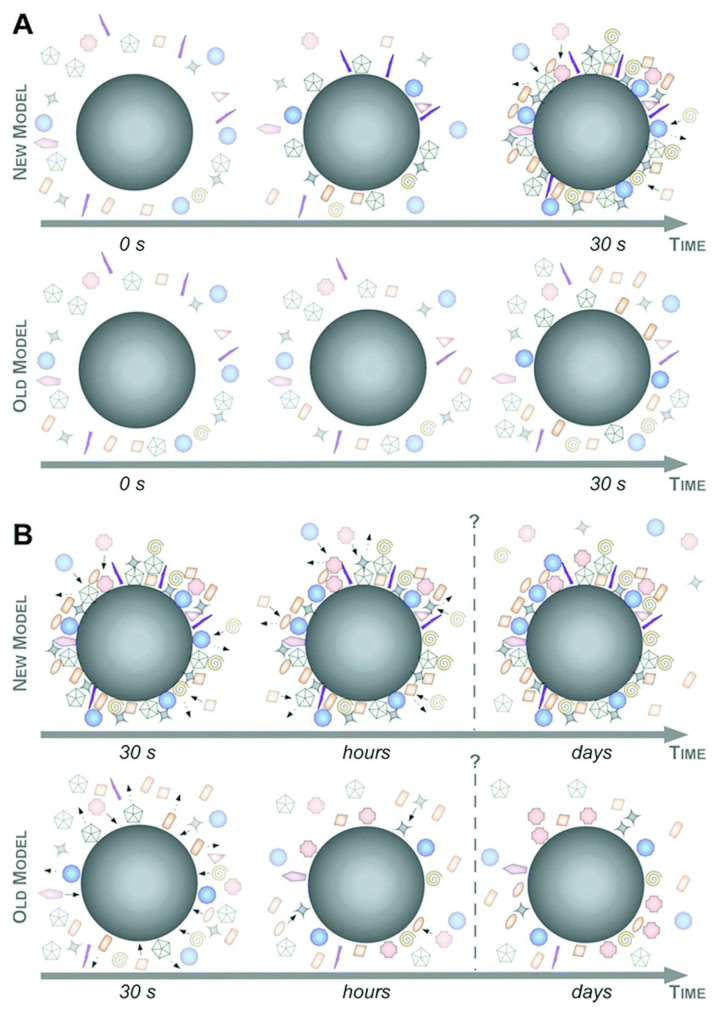
The two models of biomolecule corona evolution at the early (**A**) and the late phase (**B**). Adapted from ref. [23] with permission from The Royal Society of Chemistry.

**Figure 3 nanomaterials-14-00823-f003:**
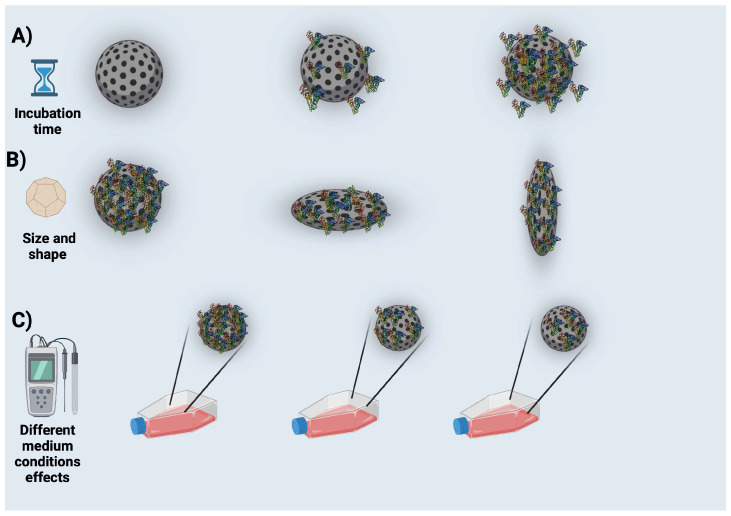
Some factors affecting protein corona (PCs). (**A**) Incubation time, (**B**) size and shape, (**C**) different medium conditions. Created with BioRender.com.

**Figure 4 nanomaterials-14-00823-f004:**
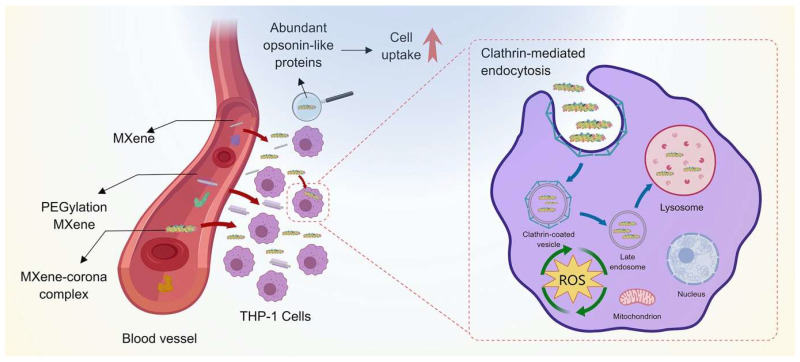
Demonstration of cellular uptake and cytotoxicity of PEGylated MXene nanostructure mediated by the protein corona. ROS: reactive oxygen species. Reprinted with permission from [65]. Copyright © 2024 Elsevier.

**Figure 5 nanomaterials-14-00823-f005:**
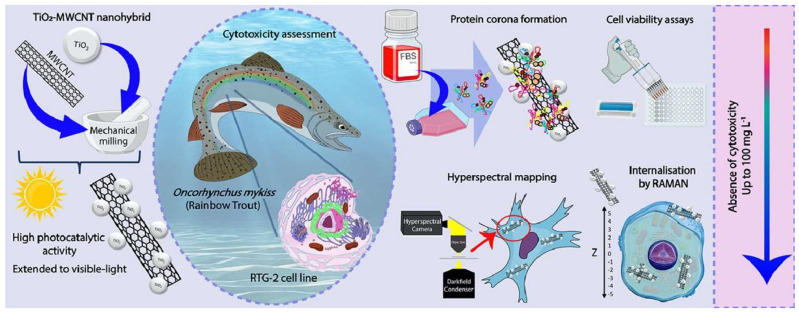
Cytotoxicity, protein corona formation, and cellular internalization of TiO_2_—MWCNT nanohybrid structures in gonadal rainbow trout tissue cells (RTG-2). Reprinted with permission from [72]. Copyright © 2024 Elsevier.

**Figure 6 nanomaterials-14-00823-f006:**
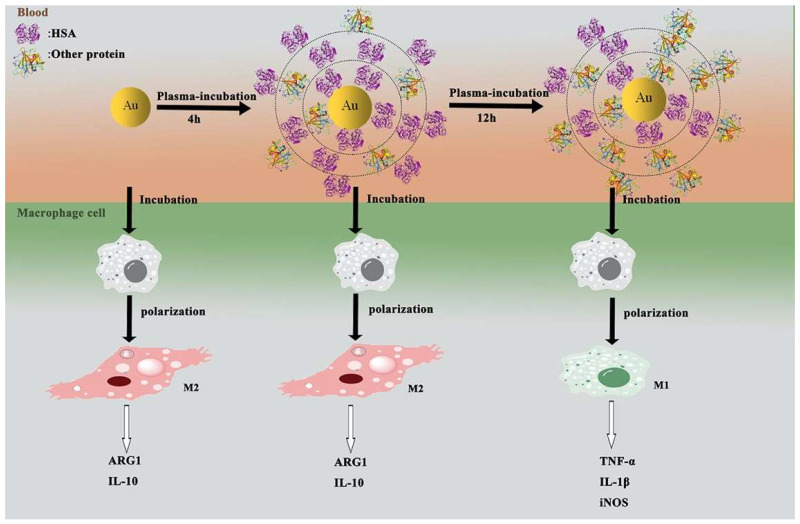
The evolution of gold (Au) nanoparticle protein coronas is affecting macrophage polarization. ARG-1: arginase 1; IL-10: interleukin-10; TNF-α: tumor necrosis factor α; IL1-β: interleukin-1β; iNOS: inducible NO synthase. Reprinted with permission from [92]. Copyright © 2024 Elsevier.

**Figure 7 nanomaterials-14-00823-f007:**
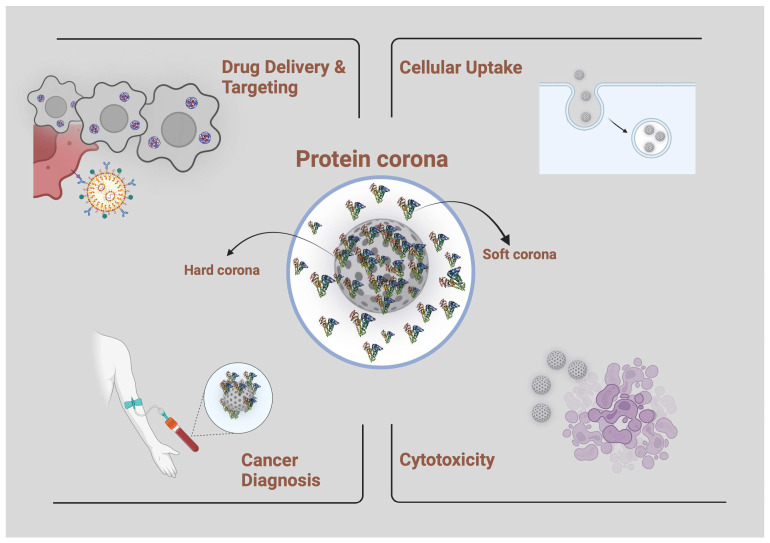
Relevant areas of nanomedicine for a targeted exploitation of the protein corona. Created with BioRender.com.

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
