# Peer review of "A Review for Uncovering the “Protein-Nanoparticle Alliance”: Implications of the Protein Corona for Biomedical Applications"

_nanomaterials, 2024, doi:10.3390/nano14100823_

Round 1

Reviewer 1 Report

Comments and Suggestions for Authors

This work summarized the protein corona around the nanoparticles for biomedical applications. This work was not well organized and major issues were as following:

(1)    What was the main purpose of this summary? The main opinions of this review were not clear.

(2)    The frame of this review was not clear.

(3)    More recent and important references should be summarized.

(4)    Representative Figures were not enough.

(5)    The article does not provide a comprehensive overview of the content involved in the topic.

Author Response

Reviewer 1

“This work summarized the protein corona around the nanoparticles for biomedical applications. This work was not well organized and major issues were as following:...”

Answer: We fully value the reviewer´s time and expertise, followed the reviewer’s suggestions, addressed all comments, and modified the revised manuscript accordingly (see detailed response below).

“1. What was the main purpose of this summary? The main opinions of this review were not clear.

  1. The frame of this review was not clear.
  2. The article does not provide a comprehensive overview of the content involved in the topic.

Answer: We apologize to the reviewer for this lack of clarity. Thus, we completely revised all sections of the manuscript to clarify the purpose of our review, as well as the term “protein-NP alliance”. Additionally, we apologize to authors whose relevant publications were not cited due to space limitations.

e.g.

Interestingly, recent endeavors have been undertaken to harness the potential of the protein corona instead of evading its presence. Exploitation of this ‘protein-nanoparticle alliance’ has significant potential to change the field of nanomedicine. Here, we present a thorough examination of the latest research on protein corona, encompassing its formation, dynamics, recent developments, and diverse bioapplications. Furthermore, we also aim to explore the interactions at the nano-bio interface, paving the way for innovative strategies to advance the application potential of the protein corona...

In this review, we aim to compile the latest advances in the understanding of PC formation and structure, as well as to highlight recent innovations in biomedical applications. We also focus on the ‘protein-NP alliance’ showcasing the key points and potential avenues for further advancement of PC exploitation. Key mechanisms, problems, and promises of controlling the protein corona are discussed.

“3. More recent and important references should be summarized.”

Answer: We value this important remark and completely revised all references. On average, approximately 30-35% of the references in articles pertain to the last four years which is in line with typical journal guidelines.

“4. Representative Figures were not enough.”

Answer: We value the referee´s remark increasing interest for the readership, and added a new Figure 1 to explain the concept of the “protein-NP alliance” and Figure 1 was changed to summarizing Figure 7. Due to limited space and typical formatting requirements of journals, we prefer not to exceed the number of figures further.

Figure 1. Two complementary concepts of understanding and working with the protein corona formed around nanoparticles (NPs). The ‘classical’ stealth coating of NPs primarily represents a ‘camouflaging technique’ seeking to ensure evasion of the immune response.Additionally, understanding this system as a beneficial ‘protein-NP alliance’, opens up new possibilities for utilizing and fine-tuning the naturally formed PC of NPs. PEG, polyethylene glycol; POx, poly(2-oxazoline).

Figure 7. Relevant areas of nanomedicine for a targeted exploitation of the protein corona. Created with BioRender.com.

In summary, we are highly thankful for the constructive review of our manuscript, which helped us to improve the quality of the revised manuscript. By modifying the manuscript accordingly, we hope that the revised manuscript is now considered acceptable by reviewer #1 for publication in Nanomaterials.

Reviewer 2 Report

Comments and Suggestions for Authors

As nanomedicine is actively developing, the authors’ review is of great interest. Nanoparticles have key benefits as novel pharmaceuticals. They can be used for drug delivery and as effector particles. The surface of the nanoparticles largely determines their properties. In environment and any biological fluid, complexes will form between nanoparticles and proteins. Studying the mechanisms of formation and effects of protein corona are of great importance. Interestingly, the corona can be controlled by changing pH, for example, or temperature. In fact, this is not just a complex, but a real alliance, the effects of which have yet to be studied. The authors provided a high-level review, which contains a large number of references from the last 25 years.

However, I had a few minor comments while reading. I hope the authors will take them into account.

1. It is not very clear what the lower right part of Figure 1 illustrates (which shows the complex of an effector T lymphocyte with a cancer cell involving interferons). Where does the nanoparticle with a protein crown fit in?

2. Moreover, I advise you to remake Figure 1 as a summary of the review (not only Introduction) and place it at the end of the article as a summarizing scheme.

3. Please expand the caption for the Fig. 2; it is not clear what is shown in panels A and B.

4. I did not find any references to figures 1, 3, 4, 5, 6 in the text.

5. In the caption for Fig. 4, decrypt PC and ROS, please. The same is for Figures 5 and 6. In fact, it would be nice to give the decoding of ARG1, TNF, iNOS for Fig. 6.

6. In line 300, the abbreviation REE is given without decoding and appears only once. Does it make sense? The same goes for the BBB abbreviation. You only use it twice (lines 354 and 494). It seems to me that unnecessary abbreviations make it difficult to read the text.

Author Response

Reviewer #2

“As nanomedicine is actively developing, the authors’ review is of great interest. Nanoparticles have key benefits as novel pharmaceuticals. They can be used for drug delivery and as effector particles. The surface of the nanoparticles largely determines their properties. In environment and any biological fluid, complexes will form between nanoparticles and proteins. Studying the mechanisms of formation and effects of protein corona are of great importance. Interestingly, the corona can be controlled by changing pH, for example, or temperature. In fact, this is not just a complex, but a real alliance, the effects of which have yet to be studied. The authors provided a high-level review, which contains a large number of references from the last 25 years.

However, I had a few minor comments while reading. I hope the authors will take them into account.”

Answer: We are pleased to learn that the referee considers our work as highly relevant for the field. We fully value his/her time and expertise, followed the reviewer’s suggestions, addressed all comments, and modified the revised manuscript accordingly (see detailed response below).

“1. It is not very clear what the lower right part of Figure 1 illustrates (which shows the complex of an effector T lymphocyte with a cancer cell involving interferons). Where does the nanoparticle with a protein crown fit in?”

Answer: We value this important remark and revised Figure 1 (now Figure 7) to increase comprehensibility. To provide a simple overview of examples where targeted exploitation of the protein corona plays a role, we chose a simplified illustration.

Figure 7. Relevant areas of nanomedicine for a targeted exploitation of the protein corona. Created with BioRender.com.

“2. Moreover, I advise you to remake Figure 1 as a summary of the review (not only Introduction) and place it at the end of the article as a summarizing scheme.”

Answer: Great suggestion! We revised Figure 1 and arranged it as summarizing Figure 7 at the end of the manuscript according to your suggestions. In addition,  we added a new figure (Figure 1) to illustrate the concept of the “protein-NP alliance”.

Figure 1. Two complementary concepts of understanding and working with the protein corona formed around nanoparticles (NPs). The ‘classical’ stealth coating of NPs primarily represents a ‘camouflaging technique’ seeking to ensure evasion of the immune response.Additionally, understanding this system as a beneficial ‘protein-NP alliance’, opens up new possibilities for utilizing and fine-tuning the naturally formed PC of NPs. PEG, polyethylene glycol; POx, poly(2-oxazoline).

“3. Please expand the caption for the Fig. 2; it is not clear what is shown in panels A and B.

  1. In the caption for Fig. 4, decrypt PC and ROS, please. The same is for Figures 5 and 6. In fact, it would be nice to give the decoding of ARG1, TNF, iNOS for Fig. 6.

Answer: We apologize for not including the complete information in the figure legends, and changed all legends accordingly.

Figure 2. The two models of biomolecule corona evolution at the early (A) and the late phase (B). Adapted from ref. [20] with permission from The Royal Society of Chemistry.

Figure 4. Demonstration of cellular uptake and cytotoxicity of PEGylated MXene     nanostructures mediated by the protein corona. ROS: reactive oxygen species. Reprinted with permission from [46]. Copyright © 2024 Elsevier.

Figure 5. Cytotoxicity, protein corona formation, and cellular internalization of TiO2−MWCNT nanohybrid structures in gonadal rainbow trout tissue cells (RTG-2). Reprinted with permission from [53]. Copyright © 2024 Elsevier.

Figure 6. The evolution of gold (Au) nanoparticle protein coronas is affecting macrophage polarization. ARG-1: arginase 1; IL-10: interleukin-10; TNF-a: tumor necrosis factor a; IL1-b: interleukin-1b; iNOS: inducible NO synthase. Reprinted with permission from [73]. Copyright © 2024 Elsevier.

  1. I did not find any references to figures 1, 3, 4, 5, 6 in the text.

Answer: We thank the reviewer for this comment, and added references to all figures in the text.

  1. In line 300, the abbreviation REE is given without decoding and appears only once. Does it make sense? The same goes for the BBB abbreviation. You only use it twice (lines 354 and 494). It seems to me that unnecessary abbreviations make it difficult to read the text.

Answer: We agree with this comment, and removed the unnecessary abbreviation (BBB). We kept REE to abbreviate “rare earth elements”, and replaced it in the relevant section.

The potential release of rare earth elements (REE) into the environment and their subsequent ingestion by humans has raised concerns due to their widespread use. Therefore, it is crucial to assess the cytotoxicity of these elements. Feng et al. conducted a study investigating the effects of three typical REE ions (La, Gd, and Yb) as well as nanometer- and micrometer-sized oxides on red blood cells, which are potential targets when introduced into the bloodstream. They examined hemolysis induced by REE at concentrations ranging from 50 to 2000 μmol/L to simulate cytotoxicity under medical or occupational exposure scenarios. The study revealed that the occurrence of hemolysis due to REE exposure is significantly dependent on their concentration. Furthermore, the observed cytotoxic effects followed a specific order, with La3+having the highest impact, followed by Gd3+, and then Yb3+. Interestingly, while the cytotoxicity of REE ions exceeded that of rare earth element oxides (REOs), nanometer-sized REOs induced more hemolysis than micrometer-sized ones. Investigations into the mechanism showed that REE caused cell membrane rupture through reactive oxygen species (ROS)-associated chemical oxidation, as evidenced by the generation of ROS, ROS quenching assays, and detection of lipid peroxidation. Additionally, it was suggested that the formation of a protein corona on REE increases steric repulsions between these elements and cell membranes, thereby reducing their cytotoxicity [51].

In summary, we are extremely grateful for the constructive review of our manuscript, which helped us improve the quality of the revised manuscript. By modifying the manuscript accordingly, we hope that the revised manuscript will now be considered acceptable for publication in Nanomaterials by reviewer #2.

Reviewer 3 Report

Comments and Suggestions for Authors

The review is dedicated to the interesting topic of protein-nanoparticle interactions and the consequences of protein corona formation. However, despite a lot of information about different studies of such interactions, I did not find any analysis of it. To begin with, the description of the Vroman effect is not clear. What about the size, mobility and affinity of the proteins? Mathematical models to describe the adsorption of blood proteins? Next, pH effects, salt effects, etc. depend on protein structure and charge, ionic strength of the solution, salt content, surface charge of the nanoparticle, and many others. There is a lot of data on this in the literature, but nothing in this review. The most relevant techniques for studying nanoparticles are electron microscopy, atomic force microscopy, dynamic light scattering, X-ray and neutron scattering. Again, nothing about these methods and their results in elucidating the main physicochemical principles of proterin-NP interactions and protein corona formation in different media. Without these data, the purpose of the review is unclear. 

Comments on the Quality of English Language

Please check the grammar and correct any spelling mistakes in the text.

Author Response

Reviewer 3

“The review is dedicated to the interesting topic of protein-nanoparticle interactions and the consequences of protein corona formation. However, despite a lot of information about different studies of such interactions, I did not find any analysis of it.”

Answer: We are pleased to learn that the referee considers our work as highly relevant for the field. We fully value his/her time and expertise, followed the reviewer’s suggestions, addressed all comments, and modified the revised manuscript accordingly (see detailed response below).

“To begin with, the description of the Vroman effect is not clear. What about the size, mobility and affinity of the proteins? Mathematical models to describe the adsorption of blood proteins?”

Answer: We value this important remark and revised the manuscript to increase comprehensibility, especially of the Vroman effect (also shown in Figure 2). Concerning mathematical models, we refer to other, more specialized literature.

The adsorption and desorption of proteins within the protein corona are continuously taking place, primarily regulated by a phenomenon known as the ‘Vroman effect’ [19]. There are two models of the Vroman effect described in the literature (Figure 2). For the 'new' approach, a highly complex corona, which can consist of numerous core-shell structures for the early stage of corona formation, is established already within 30 seconds (Figure 2A, upper panel). In contrast, the 'old' model assumes slowly evolution of low-complexity corona (Figure 2A, lower panel). In the late stage described by the ‘new’ model, the corona composition is relatively constant ex situ and due to the Vroman effect, and changes predominantly quantitatively rather than qualitatively (Figure 2B, upper panel). The PC undergoes substantial changes over time in the 'old' approach, driven by the 'Vroman effect', resulting in a highly dynamic system (Figure 2B, lower panel) [20]…. Regardless of the specific dynamics of PC formation, it is largely dependent on the physicochemical properties of the NPs, such as surface, size, and shape, as well as the surrounding biological and physicochemical environment [11]. There are several studies that attempt to model the absorption of proteins onto various biomaterials, including NPs, using mathematical models, thus predicting changes in their biological identity.

“Next, pH effects, salt effects, etc. depend on protein structure and charge, ionic strength of the solution, salt content, surface charge of the nanoparticle, and many others. There is a lot of data on this in the literature, but nothing in this review. The most relevant techniques for studying nanoparticles are electron microscopy, atomic force microscopy, dynamic light scattering, X-ray and neutron scattering. Again, nothing about these methods and their results in elucidating the main physicochemical principles of protein-NP interactions and protein corona formation in different media. Without these data, the purpose of the review is unclear.”

Answer: We greatly appreciate the referee's suggestions. However, it's essential to underscore the core focus of this review, which delves into the behavior of nanostructures, encompassing biological entities like exosomes and viruses, within diverse biological environments, and the critical role of secondary interactions. Our aim is to synthesize recent discoveries and anticipate emerging trends in protein corona-related research. Regarding your suggestion, we have opted for a distinctive approach, shifting the emphasis from the "nano" to the "bio" aspect at the nano-bio-interface. However, we added reference to other work providing more details on bio-physical techniques for protein-NP interaction.

Upon taking the body, nanoparticles (NPs) have the potential to experience a range of transformations, including protein adsorption, dissolution, agglomeration/aggregation, structural deformities, and redox reactions. These alterations play a crucial role in influencing the absorption, availability, movement, and ultimate destiny of NPs, which in turn dictate their effectiveness in therapy, diagnostic accuracy, or potential toxicity [43].

“Please check the grammar and correct any spelling mistakes in the text.”

Answer: The entire manuscript has been thoroughly revised and checked for formal errors with the assistance of a professional service.

In summary, we are extremely grateful for the constructive review of our manuscript, which helped us improve the quality of the revised manuscript. By modifying the manuscript accordingly, we hope that the revised manuscript will now be considered acceptable by reviewer #3 for publication in Nanomaterials.

Reviewer 4 Report

Comments and Suggestions for Authors

The paper is devoted to the description of current state-of-the-art in the area of protein corona on the surface of nanoparticles used for biomedical applications. Despite the fact that the topic of the review is of great interest to all specialists involved in drug delivery systems and in general the application of nanoparticles in biomedicine, this review does not fulfill its purpose and cannot be published as presented.

General comments

1. The authors have not defined what is “protein-nanoparticle alliance”? It is hard to understand what they are analyzing and discussing, without such definition.

2. The authors have not paid attention to the composition of the protein corona and the effect on it of charge (zeta potential), chemical composition, and, obviously importantly, the elasticity of the nanoparticles.

3. The authors are completely unspecific in their discussions and conclusions. They use general words, which in most cases are already obvious. For example: Page 3, line 99: “For the 'old' approach, the highly dynamic PC changes significantly over time, controlled by the 'Vroman effect.” Significantly? What does it mean?

Page 4, line 150: “The roughness of surface significantly reduces the repulsive interactions, leading to a notable impact on the quantity of proteins while leaving their identity unaffected” Notable? It does not explain how! What is their identity?

Page 5, line 162: “The human body contains a multitude of dynamic mechanisms that can effectively stabilize, solubilize, and modify particles through biomacromolecules, changes in ionic strength, alterations in pH, and active biological processes”. This could be stated without reading any papers. What are these effects and their mechanisms?

Page 5, line 193: “The composition of PCs can vary significantly when NPs are exposed to BFs in vivo or in vitro [43]. It has been proven that the protein composition in the PC is significantly influenced by both static and dynamic conditions [44].” How it significantly influenced by such conditions? What is the effect?!

The list could be continued…..

 Thus, I can conclude that authors do not provide proper analysis of the literature, which revealing specific mechanisms, compositions, effects, but simply talk about what affects. This devalues the review for readers. Review should provide specific information and analysis, which helps the researchers in understanding and planning of further studies. This review doesn’t contain such info or analysis.

4. The nanoparticulate systems reviewed non-systematically, which makes it impossible to catch any general idea. The choice of NPs is sometimes very strange. Why authors decided to review pharmacologically unimportant particles?

5. Figures should be discussed in the text of the review. However, I failed to find such discussions.

Specific comments

1. Page 1, Line 35: I would talk in this case about low «bioavailability»

2. Page 2, lines 64-68: The sentence is unclear and hard to get its sense.

3. Page 3, line 83: “Who is how important in this alliance?” May be “what is how….”?

4. Page 3, line 107, Figure 2. (A) and (B) are missing in the caption to the figure. Please add the explanation.

5. Page 4, line 110: “Effect the surface of NPs” I would propose to change to “Effect of the NPs surface area”. In general authors talk about the surface area in this paragraph.

6. Page 4, lines 117-121: This description requires better explanation. The conclusion is unclear “The polystyrene nanoparticles displayed a contrasting pattern in their behavior”. Ok. So what?!

7. Page 4, line 119: “…discovered that their ability to bind was reduced in the presence…” To bind what?!

8. Page 4, line 123: “The size and shape”. These parameters are determine surface area and surface energy, so this paragraph should be united with previous one.

9. Page 4, line 133: “Conversely, smaller NPs display an enhanced surface curvature, leading to a reduction in conformational alterations, advancement of constituents, and higher levels of adsorbed proteins”. How conformational alterations affected by larger surface curvature and greater surface area?

10. Page 4, lines 137-152: The part is titled as “The biological environment”, but contains the discussion of particles charge and hydrophobicity. Meanwhile, the role of hydrophobicity and charge is not revealed at all. Specific examples of hydrophobic and hydrophilic particles are not given. How can hydrophobic particles denature proteins? The effect of charge is mentioned, but it is not disclosed how positive or negative charge affects the composition of the protein corona. What specific proteins are included in its composition in this or that case? As noted above, the authors say nothing about the softness or stiffness of the particles, which is already cited in many modern reviews as an important factor affecting corona formation. Thus, this part not only does not correspond to its title in content, but also contains scanty unsystematized information.

11. Page 5, line 177: “After entering the physiological medium, which includes blood, interstitial fluid and intracellular environment, various forces mainly exist in the interaction of NPs with biomolecules, including van der Waals forces, hydrogen bonds, electrostatic, hydrophilic/hydrophobic interactions. These forces are attributed to the high surface energy and distinctive surface chemistry of the NPs. As a result, it subsequently gives rise to the occurring of a biomolecular corona structure”. This part is analytically very poor. Authors just mentioned possible types of noncovalent interactions (forgetting about ionic interactions) and state that these forces attribute to the formation of protein corona. It's an obvious statement that doesn't make any sense. Better analysis should be provided.

12. Page 5, line 183: “The physicochemical environment”. How authors can separate biological and physicochemical environment? Any biological environment possess certain physicochemical characteristics. Line 187: “The effect of temperature on protein diffusion and their affinity for NPs can be observed even in the physiological range of approximately 37 to 40°C/41°C [41].” What is the effect and what id the difference for PC?!

13. Page 6, Line 215: What is “MXene Ti3C2Tx”? Why someone need to introduce them into the body? PEGylation effect is known for decades, and its mechanism is well-known. Why didn’t authors mentioned that fact together with appropriate referencing? Line 217: “They presented the possibility that the PEGylation process could change the interactions between Ti3C2Tx and serum proteins and caused a major transformation in the fingerprint of the PC.” What are these changes??? Line 223: “And also, Ti3C2Tx has a dual effect as both a stimulator and scavenger of ROS in THP-1 cells, which are affected by PEGylation and PC formation.” According to grammar, you have said that cells are affected by PEGylation. Is this right?

14. Page 7, line 230: Please explain what is “cellular destiny of the nanocarriers” within the text of the paper.

15. Page 7, line 238: “Ultimately, their findings indicate that irrespective of the formulation of the NPs, the uptake of these particles by model cancer and endothelial cells decreases when they are pre-coated at higher temperatures or plasma concentrations” So what? How this could be used? May be this specific only for this type of particles?

16. Page 7, line 255: “The investigation of the internalization mechanism of AuNP uptake involved conducting studies while employing inhibitors for clathrin- and caveolin-mediated endocytosis.” What these studies showed? How protein corona affect will it be clathrin- or caveolin-mediated endocytosis? Line 258: “These findings indicate that under these circumstances, the primary mechanism accountable for the cellular uptake is the aggregation of NP” NPs aggregation could not be the mechanism of uptake but could only affect the uptake!

17. Page 8-9, “Decreasing Cytotoxicity”. The choice of particles is unclear. There are many data for other types of particles, which possess greater pharmacologically active potential that chosen by authors. Why were they not considered by authors? Figure 5 was not discussed in the text.

18. Page 9, “Improving Drug Delivery and Targeting”. Line 340: “DSPG sLip” should be clarified at first appearance in the text. In general there is a very poor explanation of the mechanism by which PC could affect efficacy of DD.

19. Page 10, lines 364-383: There are too many statements and only one reference.

20. Page 10, line 392: “Prior to injection, the formation of the desired PC around NPs can be achieved, or NPs are able to be functionalized to nucleate in vivo a PC of de-opsonin, such as albumin, transferrin, or ApoE.” The principles of how PC should be chosen should be clarified.

22. Page 11, line 400: “Making progress in cancer diagnosis” It should be clarified, that authors are talking about in vitro diagnostics.

23. Page 11, line 421: “Consequently, the presence of certain proteins (such as Integrin beta-2, Lactotransferrin, etc.) allows for the sensitive differentiation of PC from meningeal tumor patients and healthy individuals.” It is not clear how it was reached? Please explain this matter in the text of the paper.

24. Page 11, line 440: “Macrophages act a important role in the immune system and can undergo polarization into either a pro-inflammatory (M1) or anti-inflammatory (M2) phenotype.” First, “act a important role” is a grammar mistake. Second, the statement given without reference.

25. Figure 6 was not explained or discussed somehow in the text.

26. Page 12, line 462: “However, the detailed mechanism of PC formation on exosomes are not fully understood.” Mechanism – are….

27. Page 13, line 487: “When studies on exosomes are thoroughly investigated, it can be seen that…” Studies were investigated…

28. Page 13, line 493: “Compared to synthetic nanomaterials, exosomes, being natural nanomaterials, possess significant and distinctive benefits.” What are these benefits? No clarification provided

29. Page 13, line 519: “The administration of oncolytic adenoviruses through intravenous means holds great promise as a method for treating tumors.” No reference was provided to prove this statement.

30. Page 13, line 528: “After identifying the essential protein components of the virus-PC, a virus-PC replacement method was applied in which an artificial virus-PC was created on oncolytic adenoviruses to completely block the interaction of oncolytic adenoviruses with the main virus-PC components in plasma.” It would be interesting and important to read about “essential protein components” within THIS review.

31. Page 13, line 532: “The study here appeared to significantly increase the circulation time…” The study increased the circulation time??

32. Page 14, lines 544-548: “Consequently, these alterations have an impact on the biological destiny of NPs, encompassing their biodistribution, pharmacokinetics, and therapeutic effectiveness. It is widely recognized that slight variations in the composition of a protein-containing fluid, such as plasma and serum, can significantly alter the structure of the protein corona that develops on the surface of NPs.” This part was poorly discussed in the paper. Better to say – not discussed at all.

 33. Papers 2020-2024 – 32%. It is a good parameter.

Comments on the Quality of English Language

The text contains quite a few grammatical errors. The writing style often makes it impossible to understand the meaning.

Author Response

Reviewer 4

“The paper is devoted to the description of current state-of-the-art in the area of protein corona on the surface of nanoparticles used for biomedical applications. Despite the fact that the topic of the review is of great interest to all specialists involved in drug delivery systems and in general the application of nanoparticles in biomedicine, this review does not fulfill its purpose and cannot be published as presented.”

Answer: We are pleased to learn that the referee considers our work as highly relevant for the field. We fully value his/her time and expertise, followed the reviewer’s suggestions, addressed all comments, and modified the revised manuscript accordingly (see detailed response below).

General comments

  1. The authors have not defined what is “protein-nanoparticle alliance”? It is hard to understand what they are analyzing and discussing, without such definition.

Answer: We apologize to the reviewer for this lack of clarity. Thus, we completely revised all sections of the manuscript to clarify the purpose of our review, as well as the term “protein-NP alliance”, and added an additional figure (Figure 1) illustrating the concept.

Abstract: ...Interestingly, recent endeavors have been undertaken to harness the potential of the protein corona instead of evading its presence. Exploitation of this ‘protein-nanoparticle alliance’ has significant potential to change the field of nanomedicine....

...

The formation of PCs around NPs thus has practical consequences for the use of nanoparticulate tools in living systems (Figure 1). The ‘classical’ stealth coating of NPs, especially by modifications with polymers (polyethylene glycol, PEG or poly(2-oxazoline), POx) primarily aims at preventing the formation of a PC to increase half-life of the NPs by escaping the immune response (Figure 1, left). However, as the awareness of the correlation between PC profiles and the in vivo performance of NPs rises, recent endeavors have been undertaken to harness the potential of the PC, rather than evading its presence (Figure 1, right)[16]. Several preclinical studies provide evidence that the PC has the potential to facilitate the delivery of drugs to specific organs [17]. Thus, the protein corona now demands renewed focus as a prospective ally in enhancing the efficacy of NP-mediated drug delivery and turned from a foe to a friend [18]. Understanding and thus exploiting this ‘protein-NP alliance’ has significant potential to change the nanomedical field. It should be noted that both concepts shown in Figure 7 are not sharply separable and can complement each other.

Figure 1. Two complementary concepts of understanding and working with the protein corona formed around nanoparticles (NPs). The ‘classical’ stealth coating of NPs primarily represents a ‘camouflaging technique’ seeking to ensure evasion of the immune response. Additionally, understanding this system as a beneficial ‘protein-NP alliance’, opens up new possibilities for utilizing and fine-tuning the naturally formed PC of NPs. PEG, polyethylene glycol; POx, poly(2-oxazoline).

….

The ‘protein-nanoparticle alliance’, arising from this specific interaction between proteins and NPs, proves highly beneficial in numerous biomedical contexts (Figure 7). Understanding this system as an advantageous ‘protein-NP alliance’, opens up new possibilities for utilizing and fine-tuning the naturally formed PC of NPs. As we discussed here, the influence of the PC on various areas of nanomedicine is coming into focus, such as drug delivery and specific targeting, altered cellular uptake and reduced cytotoxicity. Furthermore, there are promising approaches to use PC-covered NPs in context of a ‘liquid biopsy’ to improve cancer diagnosis.

  1. The authors have not paid attention to the composition of the protein corona and the effect on it of charge (zeta potential), chemical composition, and, obviously importantly, the elasticity of the nanoparticles.

Answer: We thank the referee for this remark. To maintain the focus of our article, we referred to more specialized reviews on the topic.

Moreover, NP elasticity has also been reported in the literature to be crucial in the physiological fate of NPs, but how this occurs remains largely unknown. Li et al. revealed the mechanisms by which NP flexibility affects the physiological fate of NPs and reported that NP flexibility is an easily tunable parameter in the future rational use of PC [26].

  1. The authors are completely unspecific in their discussions and conclusions. They use general words, which in most cases are already obvious. For example: Page 3, line 99: “For the 'old' approach, the highly dynamic PC changes significantly over time, controlled by the 'Vroman effect.” Significantly? What does it mean?

Answer: We are grateful for your important comments. All sentences labeled yellow have been changed in the manuscript and are as follows:

  • Page 3, line 99: “For the 'old' approach, the highly dynamic PC changes significantly over time, controlled by the 'Vroman effect.” Significantly? What does it mean?

Answer:

The PC undergoes substantial changes over time in the 'old' approach, driven by the 'Vroman effect', resulting in a highly dynamic system.

  • Page 4, line 150: “The roughness of surface significantly reduces the repulsive interactions, leading to a notable impact on the quantity of proteins while leaving their identity unaffected” Notable? It does not explain how! What is their identity?

Answer: As a result, while the amount of protein accumulated/adsorbed on the surface of the nanostructures changes, it does not cause any effect on the structure and character of the nanostructures.

  • Page 5, line 162: “The human body contains a multitude of dynamic mechanisms that can effectively stabilize, solubilize, and modify particles through biomacromolecules, changes in ionic strength, alterations in pH, and active biological processes”. This could be stated without reading any papers. What are these effects and their mechanisms?

Answer: The human body contains a multitude of regulatory mechanisms that can effectively stabilize, solubilize, and modify particles through biomacromolecules inducing changes in ionic strength and pH [39].  After entering the physiological medium, which includes blood, interstitial fluid, and intracellular environment, various forces mainly exist in the interaction of NPs with biomolecules, including van der Waals forces, hydrogen bonds, electrostatic, and hydrophilic/hydrophobic interactions. These forces are attributed to the high surface energy and distinctive surface chemistry of the NPs. As a result, it subsequently gives rise to the development of a biomolecular corona structure [40].

  • Page 5, line 193: “The composition of PCs can vary significantly when NPs are exposed to BFs in vivo or in vitro [43]. It has been proven that the protein composition in the PC is significantly influenced by both static and dynamic conditions [44].” How it significantly influenced by such conditions? What is the effect?!

Answer: It has been proven that the protein composition in the PC is significantly influenced by both static conditions, such as flow rate, and dynamic conditions. The PC exhibits less homogeneity during dynamic circumstances, allowing certain NPs to remain uncoated and available for interaction with cells. [44].

  1. The nanoparticulate systems reviewed non-systematically, which makes it impossible to catch any general idea. The choice of NPs is sometimes very strange. Why authors decided to review pharmacologically unimportant particles?

Answer:  Thank you for your valuable feedback. While the selection of nanoparticles is indeed a complex issue, we aimed to provide a general evaluation based on their surface properties, sizes, and characteristics. It is well known that each nanostructure may possess unique properties, and our intention was to capture this diversity. However, we understand your concern and acknowledge that some of the nanoparticles reviewed may not seem pharmacologically relevant.

  1. Figures should be discussed in the text of the review. However, I failed to find such discussions.

Answer: We thank the reviewer for this comment, and added references to all figures in the text.

Specific comments

  1. Page 1, Line 35: I would talk in this case about low «bioavailability»

Answer: Thank you for your siginificant contribution. This information represents an assessment obtained from the cited literature rather than our own private interpretation.

  1. Page 2, lines 64-68: The sentence is unclear and hard to get its sense.

The composition of PC exhibits notable variations depending on the sources and/or fabrication methods employed for NP systems [11]. For example, the method of gold NPs production had an impact on their specific biomolecule binding profiles (albumin, RNA) [15].

  1. Page 3, line 83: “Who is how important in this alliance?” May be “what is how….”?
  2. What is how important in the ‘protein-nanoparticle alliance’?

  1. Page 3, line 107, Figure 2. (A) and (B) are missing in the caption to the figure. Please add the explanation.

Answer: It was revised in the manuscript.

Figure 2. The two models of biomolecule corona evolution at the early (A) and the late phase (B). Adapted from ref. [20] with permission from The Royal Society of Chemistry.

  1. Page 4, line 110: “Effect the surface of NPs” I would propose to change to “Effect of the NPs surface area”. In general authors talk about the surface area in this paragraph.

Answer: We simplified this subheading, and changed it to “a) The NP surface”.

  1. Page 4, lines 117-121: This description requires better explanation. The conclusion is unclear “The polystyrene nanoparticles displayed a contrasting pattern in their behavior”. Ok. So what?!

Answer: An important example of this is that silica NPs have been shown to bind to more plasma proteins at lower concentrations of plasma proteins than polystyrene NPs of the same size. However, it was discovered that their ability to bind was reduced in the presence of elevated levels of plasma proteins in the surrounding environment. The polystyrene nanoparticles displayed a contrasting pattern in their behavior [28].

  1. Page 4, line 119: “…discovered that their ability to bind was reduced in the presence…” To bind what?!

Answer: However, it was discovered that their ability to bind proteins was reduced in the presence of elevated levels of plasma proteins in the surrounding environment [22].

  1. Page 4, line 123: “The size and shape”. These parameters are determine surface area and surface energy, so this paragraph should be united with previous one.

Answer: We thank the reviewer for this comment, and changed the sections accordingly.

  1. Page 4, line 133: “Conversely, smaller NPs display an enhanced surface curvature, leading to a reduction in conformational alterations, advancement of constituents, and higher levels of adsorbed proteins”. How conformational alterations affected by larger surface curvature and greater surface area?

Reply: Dear reviewer, thank you for your comments.  We added necessary additions as follows:

To elaborate, smaller particles exhibit higher surface curvature, smaller surface area, and limited surface interaction with proteins. This results in a lesser degree of protein coverage. Larger particles have lower surface curvature, larger surface area, and provide large surface interaction for proteins, thus facilitating protein coverage to a greater extent [1].

  1. Page 4, lines 137-152: The part is titled as “The biological environment”, but contains the discussion of particles charge and hydrophobicity. Meanwhile, the role of hydrophobicity and charge is not revealed at all. Specific examples of hydrophobic and hydrophilic particles are not given. How can hydrophobic particles denature proteins? The effect of charge is mentioned, but it is not disclosed how positive or negative charge affects the composition of the protein corona. What specific proteins are included in its composition in this or that case? As noted above, the authors say nothing about the softness or stiffness of the particles, which is already cited in many modern reviews as an important factor affecting corona formation. Thus, this part not only does not correspond to its title in content, but also contains scanty unsystematized information.

Answer: The section was completely revised to address all concerns.

In addition to the physicochemical properties of NPs mentioned above (Figure 3), the surrounding biological environment plays an important role in determining the potential biological impacts of nanostructures utilized in the biomedical field [40]. In physiological fluids, nanoparticles are very likely to engage in various colloidal interactions with various constituents, such as salts, sugars, and proteins. This ‘nano-bio interface’ involves dynamic physicochemical interactions, kinetic factors, and thermodynamic exchanges between the surfaces of nanostructures and biological components. The in vivo stability of nanostructures can be significantly impacted by these interactions, leading to devastating consequences.

Basic physical and chemical investigations are often carried out in controlled settings to preserve the entirety of the NP design and conjugation. Nevertheless, the ultimate determination of the biocompatibility and biodistribution of these particles relies on the alterations that take place within the particle upon exposure to physiological fluids [39]. Cell culture medium (CCM) is the prevailing environment that NPs commonly encounter under in vitro conditions. This medium typically comprises fetal calf or bovine serum, which is essential for promoting optimal cell growth. The evaluation of the performance and behavior of particle systems in simulated physiological solutions may be required, depending on the specific biomedical application. Moreover, it is crucial to characterize potential scenarios that NPs encounter when they enter the human bloodstream. This understanding plays a pivotal role in guiding the evidence-based development of particle systems for applications in humans. The human body contains a multitude of regulatory mechanisms that can effectively stabilize, solubilize, and modify particles through biomacromolecules inducing changes in ionic strength and pH [39].  After entering the physiological medium, which includes blood, interstitial fluid, and intracellular environment, various forces mainly exist in the interaction of NPs with biomolecules, including van der Waals forces, hydrogen bonds, electrostatic, and hydrophilic/hydrophobic interactions. These forces are attributed to the high surface energy and distinctive surface chemistry of the NPs. As a result, it subsequently gives rise to the development of a biomolecular corona structure [40].

  1. Page 5, line 177: “After entering the physiological medium, which includes blood, interstitial fluid and intracellular environment, various forces mainly exist in the interaction of NPs with biomolecules, including van der Waals forces, hydrogen bonds, electrostatic, hydrophilic/hydrophobic interactions. These forces are attributed to the high surface energy and distinctive surface chemistry of the NPs. As a result, it subsequently gives rise to the occurring of a biomolecular corona structure”. This part is analytically very poor. Authors just mentioned possible types of noncovalent interactions (forgetting about ionic interactions) and state that these forces attribute to the formation of protein corona. It's an obvious statement that doesn't make any sense. Better analysis should be provided.

Answer: It was revised as follows:

Comprehending the characteristics of both soft and hard coronas is of utmost importance in comprehending the stability, functionality, and interactions of nanoparticles (NPs) with biological systems. From a kinetic standpoint, the interactions between proteins and NPs in biological fluids are regulated by non-covalent forces, including electrostatic forces, hydrophobic forces, hydrogen bonding, and π-π stacking. Proteins competitively bind to the surfaces of NPs, resulting in the formation of transient NP-protein complexes that consist of both soft and hard corona proteins under thermodynamically favorable conditions. However, due to the rapid dissociation rate of soft corona proteins, the current understanding of the biological composition of the protein corona is typically limited to the hard corona proteins.

  1. Page 5, line 183: “The physicochemical environment”. How authors can separate biological and physicochemical environment? Any biological environment possess certain physicochemical characteristics.

Answer: To keep comprehensibility and a good structure of the article, we decided to separate the “biological environment” meaning primarily biomolecules, form the “physicochemical environment” characterized by physical parameters, such as pH, and temperature. Of course, both naturally belong to a common environment, but they are discussed separately here for the sake of clarity.

Line 187: “The effect of temperature on protein diffusion and their affinity for NPs can be observed even in the physiological range of approximately 37 to 40°C/41°C [41].” What is the effect and what id the difference for PC?!

Answer: Dear reviewer, thank you for your comments.  We added necessary additions as follows:

Oberländer et al. demonstrated that surface charge and surfactant composition can be influenced under conditions of constant temperature and concentration. Furthermore, they observed that the corona structure formed at low temperatures (4°C) differs from that formed at physiological temperatures (37°C). Their findings also indicated that the uptake of nanoparticles by model cancer and endothelial cells decreased with increasing temperature or plasma concentration, irrespective of nanoparticle formulation [48].

  1. Page 6, Line 215: What is “MXene Ti3C2Tx”? Why someone need to introduce them into the body? PEGylation effect is known for decades, and its mechanism is well-known. Why didn’t authors mentioned that fact together with appropriate referencing? Line 217: “They presented the possibility that the PEGylation process could change the interactions between Ti3C2Tx and serum proteins and caused a major transformation in the fingerprint of the PC.” What are these changes??? Line 223: “And also, Ti3C2Tx has a dual effect as both a stimulator and scavenger of ROS in THP-1 cells, which are affected by PEGylation and PC formation.” According to grammar, you have said that cells are affected by PEGylation. Is this right?

Reply: Dear reviewer, thank you for your valuable comments.  We added necessary additions as follows:

They presented the possibility that the PEGylation process could change the interactions between Ti3C2Tx and serum proteins, and thus cause a major transformation in the PC fingerprint. It was found that PEGylation can alter the interaction between Ti3C2Tx and serum proteins, and also, it has been reported that PEGylation and PC formation increased MXene cellular uptake.

  1. Page 7, line 230: Please explain what is “cellular destiny of the nanocarriers” within the text of the paper.

Answer: We changed the text to clarify.

…The cellular uptake of nanocarriers is influenced by alterations in the composition of the PC, which can be contingent upon certain parameters…

  1. Page 7, line 238: “Ultimately, their findings indicate that irrespective of the formulation of the NPs, the uptake of these particles by model cancer and endothelial cells decreases when they are pre-coated at higher temperatures or plasma concentrations” So what? How this could be used? May be this specific only for this type of particles?

Answer: Recent findings are summarized here to indicate that they may be also relevant for other NPs or cellular systems. We changed the text to clarify.

Although these results may be limited to the analyzed NPs, a potential influence of coating temperature and pre-incubation should be considered when aiming to deliberately alter cellular uptake of nanosystems.

  1. Page 7, line 255: “The investigation of the internalization mechanism of AuNP uptake involved conducting studies while employing inhibitors for clathrin- and caveolin-mediated endocytosis.” What these studies showed? How protein corona affect will it be clathrin- or caveolin-mediated endocytosis?

Line 258: “These findings indicate that under these circumstances, the primary mechanism accountable for the cellular uptake is the aggregation of NP” NPs aggregation could not be the mechanism of uptake but could only affect the uptake!

Answer: Thank you to the reviewer for this comment. We have revised the text for clarity

Investigations of the internalization mechanism responsible for AuNP uptake suggested that this process was not significantly influenced by either clathrin receptors or lipid rafts. These findings indicate that under cell culture conditions, the primary factor influencing cellular uptake is NP aggregation [54].

  1. Page 8-9, “Decreasing Cytotoxicity”. The choice of particles is unclear. There are many data for other types of particles, which possess greater pharmacologically active potential that chosen by authors. Why were they not considered by authors? Figure 5 was not discussed in the text.

Answer:  We apologize for this lack of clarity, and want to emphasize that our review should cover latest research mainly of the past 4 years. We included articles that investigated nanosystems relevant to applications. Of course, due to the extensive scope of research, unfortunately, not all articles could be considered.

Reference to Figure 5 was added in the text.

  1. Page 9, “Improving Drug Delivery and Targeting”. Line 340: “DSPG sLip” should be clarified at first appearance in the text. In general there is a very poor explanation of the mechanism by which PC could affect efficacy of DD.

Moreover, the strong binding affinity of anionic liposomes composed of phosphatidylglycerol (DSPG sLip) to both planktonic bacteria and biofilms was facilitated by the cumulative complement accumulation on the DSPG sLip…..

  1. Page 10, lines 364-383: There are too many statements and only one reference.

We completely agree, and added references in the section.

  1. Page 10, line 392: “Prior to injection, the formation of the desired PC around NPs can be achieved, or NPs are able to be functionalized to nucleate in vivo a PC of de-opsonin, such as albumin, transferrin, or ApoE.” The principles of how PC should be chosen should be clarified.

Answer: Thank you for this comment. We changed this paragraph to clarify.

In general, the choice of a suitable PC is depending on the intended application and function, either prolonging half-life of the NPs in a biological system, increasing specific targeting or cellular uptake. Here, more detailed studies are mandatory to reveal general principles of targeted PC design [19].

  1. Page 11, line 400: “Making progress in cancer diagnosis” It should be clarified, that authors are talking about in vitro diagnostics.

Answer: This is an important remark! We clarified this in the text.

Consequently, when NPs are incubated in these plasma samples in vivo, proteins are expected to adhere to and accumulate on them, thereby influencing the composition of their PC.

  1. Page 11, line 421: “Consequently, the presence of certain proteins (such as Integrin beta-2, Lactotransferrin, etc.) allows for the sensitive differentiation of PC from meningeal tumor patients and healthy individuals.” It is not clear how it was reached? Please explain this matter in the text of the paper.

Answer: We are happy about the reviewer´s great interest and detailed comments on the compiled studies. However, we feel that it is beyond the scope of our review article to go into details and leave it to the reader, if interested, to consult also the original work.

  1. Page 11, line 440: “Macrophages act a important role in the immune system and can undergo polarization into either a pro-inflammatory (M1) or anti-inflammatory (M2) phenotype.” First, “act a important role” is a grammar mistake. Second, the statement given without reference.

We revised the manuscript accordingly. We added a general reference, although it rather represents general scholarly opinion than a statement.

Macrophages play a crucial role in the immune system, exhibiting the ability to polarize into either a pro-inflammatory (M1) or anti-inflammatory (M2) phenotype [80].

  1. Figure 6 was not explained or discussed somehow in the text.

Answer: We thank the reviewer for this comment, and added a reference to Figure 6, as well as a detailed figure legend in the manuscript.

  1. Page 12, line 462: “However, the detailed mechanism of PC formation on exosomes are not fully understood.” Mechanism – are….
  2. Page 13, line 487: “When studies on exosomes are thoroughly investigated, it can be seen that…” Studies were investigated…

Answer: We thank you the reviwer for these important comments. This part was revised as follows in manuscript:

Given that exosomes are released by almost all cell types, they are highly abundant in BFs where they bind biomolecules, such as proteins. However, the detailed mechanism underlying the formation of PCs on exosomes remains incompletely understood. Existing literature has demonstrated PC formation around exosomes in aqueous phases, primarily through electrostatic interactions and protein aggregation [89]. However, the understanding of the factors influencing PC formation in biofluids remains both theoretically and experimentally under developed [89,90]. Previous studies have yielded conflicting results regarding the presence of albumin on the surface of exosomes [90].

  1. Page 13, line 493: “Compared to synthetic nanomaterials, exosomes, being natural nanomaterials, possess significant and distinctive benefits.” What are these benefits? No clarification provided

Reply: These benefits are listed as follows:

Compared to synthetic nanomaterials, exosomes, as natural counterparts, offer significant and distinctive advantages.  Due to their natural function as transport vehicles, exosomes are optimized in structure and composition for specific intercellular transport and cargo delivery, and thus, are not prone to be eliminated by immune response. For example, exosomes can traverse the blood-brain barrier allowing DD to the central nervous system [87–89].

  1. Page 13, line 519: “The administration of oncolytic adenoviruses through intravenous means holds great promise as a method for treating tumors.” No reference was provided to prove this statement.

Answer: We thank for this remark and added the reference accordingly.

  1. Page 13, line 528: “After identifying the essential protein components of the virus-PC, a virus-PC replacement method was applied in which an artificial virus-PC was created on oncolytic adenoviruses to completely block the interaction of oncolytic adenoviruses with the main virus-PC components in plasma.” It would be interesting and important to read about “essential protein components” within THIS review.

Answer: We completely agree, and added this detail in the text.

After identifying the essential protein components of the virus-PC, they employed a virus-PC replacement approach, creating an artificial virus-PC on oncolytic adenoviruses to block their interaction with key virus-PC components in plasma. Serum albumin (SA), fibrinogen (FB), and immunoglobulin (IM), the most common and abundant proteins in the nanoparticle PC, were found to be significantly negatively correlated with most of the major virus protein corona components identified in this study.

  1. Page 13, line 532: “The study here appeared to significantly increase the circulation time…” The study increased the circulation time??

This method significantly extended the circulation time of the adenoviruses and enhanced their distribution in tumors.

  1. Page 14, lines 544-548: “Consequently, these alterations have an impact on the biological destiny of NPs, encompassing their biodistribution, pharmacokinetics, and therapeutic effectiveness. It is widely recognized that slight variations in the composition of a protein-containing fluid, such as plasma and serum, can significantly alter the structure of the protein corona that develops on the surface of NPs.” This part was poorly discussed in the paper. Better to say – not discussed at all.

Answer: It was changed in text as follows:

Hence, comprehending the interplay between nanomaterials and biological fluids is fundamental for their utilization. The utilization of proteomics, bioinformatics, and nanotechnology in conjunction can aid in identifying the components of the PC that is created when metal NPs are exposed to biological fluids. This approach holds the potential to offer valuable insights into disease progression and facilitate the exploration of novel therapeutic targets.

In summary, we are extremely grateful for the detailed review of our manuscript, which helped us improve the quality of the revised manuscript. By modifying the manuscript accordingly, we hope that the revised manuscript will now be considered acceptable for publication in Nanomaterials by reviewer #4.

Reviewer 5 Report

Comments and Suggestions for Authors

The article talks a lot about nanoparticles in general. I think it should be specified which nanoparticles are in question - Ag, Au, ZnO....

Different nanoparticles have different structure, different properties, antibacterial activity, different specific surface area, etc.

On line 321 there is a typo of 100 mg L−1

On line 325 - TiO2 should become TiO2.

Author Response

Reviewer 5

The article talks a lot about nanoparticles in general. I think it should be specified which nanoparticles are in question - Ag, Au, ZnO....”

Different nanoparticles have different structure, different properties, antibacterial activity, different specific surface area, etc.

Answer: We are pleased that the referee evaluated our work and devoted significant time. We fully value reviewer’sexpertise, followed the reviewer's suggestions, addressed all comments, and amended the revised draft accordingly (see detailed response below). And also, "Different nanoparticles have different structures, different properties, antibacterial activity, and different specific surface areas" is a very appropriate comment. Thank you very much for this. Here, we only touched on nanoparticles in general. Therefore, as we focus on the general characteristics

-On line 321 there is a typo of 100 mg L−1

-On line 325 - TiO2 should become TiO2.

We value this important remark and revised the manuscript as ‘’ Da Silva et al. [53] synthesized titanium dioxide NPs-multi-walled carbon nanotubes (TiO2−MWCNT), and conducted a pioneering study investigating their cytotoxic effects, PC formations, and cellular uptake on fibroblasts derived from gonadal rainbow trout tissue (RGT-2) (Figure 5). Remarkably, this nanohybrid exhibited no toxicity towards RTG-2 cells even at high concentrations and 24 h of exposure’’

In summary, we are extremely grateful for the detailed review of our manuscript, which helped us improve the quality of the revised manuscript. By modifying the manuscript accordingly, we hope that the revised manuscript will now be considered acceptable for publication in Nanomaterials by reviewer #5.

Round 2

Reviewer 3 Report

Comments and Suggestions for Authors

Accept in present form

Author Response

We are extremely grateful for the detailed review of our manuscript, which helped us improve the quality of the revised manuscript. By modifying the manuscript accordingly, we hope that the revised manuscript will now be considered acceptable for publication in Nanomaterials by reviewer #3.

Reviewer 4 Report

Comments and Suggestions for Authors

Author did a great work to improve their MS. However, they didn't met my principle comments related to

(1) protein corona composition,

(2) choice of nanoparticles nature for review.

Withour such specific information this review will be not usefull for great number of researchers in the area. The review should be systematic and based on structure-property principle. It is a pity, but I didn't realized what are the proteins within the protein corona and how protein composition is differed depending on the nature of the particles. The chemistry and biology of such proteins will definetely affect the discussed properties and this effects shouldl be outlined. 

I encourage authors to revise their manuscript regarding the mentioned issues. Also I would ask to use specific references to their corrections in the text of MS within the text of their answers. I mean - Page ?, Line ? "what was done?" Otherwise it is hard to find what was done. Just yellow marker doesn't help in this matter.

Comments on the Quality of English Language

English language of the manuscript was improved. It is more or less fine.

Author Response

Reviewer 4

“Author did a great work to improve their MS. However, they didn't met my principle comments related to

(1) protein corona composition,

(2) choice of nanoparticles nature for review.

Answer: We fully value the reviewer´s time and expertise, followed the reviewer’s suggestions, addressed all comments, and modified the revised manuscript accordingly (see detailed response below).

The authors would like to thank the reviewer for the reviewer's suggestions. A new section on this subject, marked in yellow below, has been added to the article. We have already mentioned this section in the "Effects of surface, size and shape of NPs" section. However, based on this valuable comment of the reviwer, we wanted to make a little clarification on "Nanoparticles choice in protein corona". For more information, we wanted to keep the article structure and the number of characters faithful.

In summary, we are extremely grateful for the detailed review of our manuscript, which helped us improve the quality of the revised manuscript. By modifying the manuscript accordingly, we hope that the revised manuscript will now be considered acceptable for publication in Nanomaterials by reviewer #4.

(1) protein corona composition,

e) Importance of protein corona composition

The literature lacks a unanimous agreement regarding the benefits and drawbacks of protein adsorption [54]. The PC’s composition is intricate, diverse, and heavily influenced by the specific biological surroundings it interacts with upon contact with NPs, suggesting the potential for exposure-induced memory. Specific elements within the PC, like Opsonins (IgG, complement, among others), could enhance the rapid uptake of coated nanoparticles by the reticuloendothelial system [55,56]. Moreover, proteins known as dysopsonins, like albumin, have the ability to inhibit complement activation when bound to particles. This results in an extended circulation period and decreased toxicity [54]. Despite the fact that some proteins are unique to particular nanoparticles and the protein corona's composition is influenced by nanoparticle characteristics, albumin, IgG, fibrinogen, and apolipoprotein are consistently found in the majority of PC studies [57]. The PC’s composition is contingent upon the concentration and kinetic characteristics of the plasma proteins, in a time frame. Consequently, comprehending the dynamics of corona protein exchange becomes crucial as it dictates the PC's significance within the broader biological profile of nanoparticles [58]. It is important to recognize that the modification of the secondary structure of proteins, which affects the overall bioreactivity of NPs, is influenced by various factors. These factors include the surface area and flexibility of the NPs as well as the chemical properties of the absorbed proteins. NPs with curved surfaces, as opposed to planar surfaces, offer greater flexibility and a larger surface area for the adsorption of protein molecules [59]. Furthermore, the NPs' curved surface could potentially impact the protein's secondary structure, leading to irreversible alterations in certain instances [60]. To give an example, while the effect of gold NPs on the conformational changes and concentration-dependent behavior of BSA was demonstrated [61], in contrast to this situation, it was reported that no changes in the conformational changes of BSA could be detected when carbon C60 fullerenes were absorbed on the surface of NPs [62]. Furthermore, regarding this issue, it has been noted that conformational changes in the secondary structure of transferrin are irreversible when exposed to ultrasmall superparamagnetic iron oxide NPs [63].

The findings indicate that apart from the process of protein association with NPs, there exists another crucial aspect that remains inadequately elucidated, namely the interaction between proteins and the particle surface. While the significance of hydrophobic/hydrophilic and electrostatic interactions is acknowledged, a comprehensive description of these interactions is lacking. A thorough comprehension of these interactions holds the potential to enhance our understanding of the factors that impact the kinetics of protein binding. Consequently, there is a necessity to enhance analytical procedures and techniques to comprehensively elucidate the intricate processes involved in the interaction between proteins and nanoparticles [54].

(2) choice of nanoparticles nature for review.

a) Nanoparticles choice in protein corona

Even in the absence of any pathological conditions, achieving optimal biodistribution and drug delivery is challenging due to various obstacles encountered by NPs. These obstacles include physical barriers such as shear forces, protein adsorption, and rapid clearance, which collectively restrict the proportion of administered NPs that successfully reach the intended therapeutic site [27]. Disease states frequently lead to modifications in these obstacles, making them even more challenging to overcome through a universal, one-size-fits-all strategy. Moreover, the variations in biological barriers are not consistent among different diseases and can differ significantly from one patient to another. These variations can manifest at systemic, microenvironmental, and cellular levels, posing challenges in their comprehensive identification and characterization. Comprehending the biological obstacles encountered in a general sense as well as on an individual patient basis facilitates the development of highly efficient NP platforms [28].

Proteins and NPs are two important parameters in PC. When making design and selection here, it may also be useful to focus on proteins instead of NP selection, which is a very comprehensive and different issue in itself. While many parameters are involved in the PC interaction, it is extremely interesting to know that proteins will have different net charges at different medium pHs. Because instead of selecting a large number of NPs, it may be more beneficial to design PCs by focusing on proteins with known net charge at specific pHs or isoelectric pH points. Besides active electrostatic interactions, other secondary interactions are known to exist in PC. Of course, interacting with NP by adjusting the net charge of the protein will increase the probability of interaction and the desired interactions will be achieved thanks to NP-protein engineering. However, without ignoring the NP side, we have touched on some important parameters of some NPs, such as surface, size and shape, in the following sections.

Reviewer 5 Report

Comments and Suggestions for Authors

Appropriate corrections have been made. The article can be published.

Author Response

 We are extremely grateful for the detailed review of our manuscript, which helped us improve the quality of the revised manuscript. By modifying the manuscript accordingly, we hope that the revised manuscript will now be considered acceptable for publication in Nanomaterials by reviewer #5.

Round 3

Reviewer 4 Report

Comments and Suggestions for Authors

Authors have made all needed corrections. Manuscript could be accepted for publication.

Comments on the Quality of English Language

Minor editing of English language required